# Natural Compounds of *Lasia spinosa* (L.) Stem Potentiate Antidiabetic Actions by Regulating Diabetes and Diabetes-Related Biochemical and Cellular Indexes [note 1]

**DOI:** 10.3390/ph15121466

**Published:** 2022-11-25

**Authors:** Md. Mamunur Rashid, Md. Atiar Rahman, Md. Shahidul Islam, Md. Amjad Hossen, A. M. Abu Ahmed, Mirola Afroze, Alaa H. Habib, Manal M. S. Mansoury, Hend F. Alharbi, Reham M. Algheshairy, Walla Alelwani, Afnan M. Alnajeebi, Jitbanjong Tangpong, Srabonti Saha, Alaa Qadhi, Wedad Azhar

**Affiliations:** 1Department of Biochemistry and Molecular Biology, University of Chittagong, Chittagong 4331, Bangladesh; 2School of Allied Health Sciences, Walailak University, Nakhon Si Thammarat 80160, Thailand; 3Department of Pharmacy, Faculty of Science and Engineering, International Islamic University Chittagong, Chittagong 4318, Bangladesh; 4Department of Genetic Engineering and Biotechnology, University of Chittagong, Chittagong 4331, Bangladesh; 5Bangladesh Reference Institute for Chemical Measurements (BRiCM), Dr. Qudrat-e-Khuda Road (Laboratory Road), Dhanmondi, Dhaka 1205, Bangladesh; 6Department of Physiology, Faculty of Medicine, King Abdulaziz University, Jeddah 21589, Saudi Arabia; 7Department of Food Science and Human Nutrition, College of Agriculture and Veterinary Medicine, Qassim University, Buraydah 51452, Saudi Arabia; 8Department of Biochemistry, Collage of Science, University of Jeddah, Jeddah 80203, Saudi Arabia; 9Clinical Nutrition Department, Faculty of Applied Medical Sciences, Umm Al-Qura University, P.O. Box 715, Makkah 21955, Saudi Arabia

**Keywords:** *Lasia spinosa*, diabetes mellitus, AMPK, PPARγ, Methyl α-d-galactopyranoside, methyl α-d-glucopyranoside

## Abstract

Natural biometabolites of plants have been reported to be useful in chronic diseases including diabetes and associated complications. This research is aimed to investigate how the biometabolites of *Lasia spinosa* methanol stem (ME_X_LS) extract ameliorative diabetes and diabetes-related complications. ME_X_LS was examined for in vitro antioxidant and in vivo antidiabetic effects in a streptozotocin-induced diabetes model, and its chemical profiling was done by gas chromatography-mass spectrometry analysis. The results were verified by histopathological examination and in silico ligand-receptor interaction of characterized natural biometabolites with antidiabetic receptor proteins AMPK (PDB ID: 4CFH); PPARγ (PDB ID: 3G9E); and mammalian α-amylase center (PDB ID: 1PPI). The ME_X_LS was found to show a remarkable α-amylase inhibition (47.45%), strong antioxidant action, and significant (*p* < 0.05) decrease in blood glucose level, serum aspartate aminotransferase (AST), alanine aminotransferase (ALT), low-density lipoprotein (LDL), urea, uric acid, creatinine, total cholesterol, triglyceride (TG), liver glycogen, creatinine kinase (CK-MB), and lactate dehydrogenase (LDH) and increase in serum insulin, glucose tolerance, and high-density lipoprotein (HDL). Rat’s pancreas and kidney tissues were found to be partially recovered in histopathological analyses. Methyl α-d-galactopyranoside displayed the highest binding affinity with AMPK (docking score, −5.764), PPARγ (docking score, −5.218), and 1PPI (docking score, −5.615) receptors. Data suggest that the ME_X_LS may be an exciting source to potentiate antidiabetic activities affirming a cell-line study.

## 1. Introduction

One of the biggest public health issues in the Globe is type 2 diabetes mellitus (T2DM). It is a collective term for a variety of metabolic illnesses, the main feature of which is persistent hyperglycemia brought on by metabolic dysfunction linked to insulin resistance, which has terrible side effects and a dismal prognosis [1]. According to the International Diabetes Federation (IDF) data in 2019, 463 million adults (20–79 years) have been suffered from DM, and this number is projected to reach 700 million by 2045 [2]. High levels of hyperglycemia in people with DM are linked to microvascular consequences such as diabetic nephropathy, neuropathy, and retinopathy, as well as macrovascular issues like coronary artery disease, peripheral arterial disease, and stroke [3]. The main risk factors for the onset of diabetes-related disorders include oxidative stress, inflammation, and dyslipidemia. Uncontrolled hyperglycemia enhances lipid peroxidation, glucose oxidation, and nonenzymatic protein glycation in diabetics [4]. Although there are numerous conventional medications available, including metformin (MET), thiazolidinediones (TZDs), dipeptidyl peptidase 4 (DPP-4) inhibitors, sulfonylureas (SUF), glucagon-like peptide 1 (GLP-1), and sodium-glucose cotransporter-2 (SGLT-2) inhibitors, their side effects, affordability, and patience compliance are still the matter of concerns. Therefore, inevitability exists for innovative and potent antidiabetic medications that are accessible to humans and have low toxicity. Prevalence of patients who use medicinal plants for the management of diabetes mellitus is increased in recent years and the World health organization (WHO) estimated that 25% of the currently available drugs are derived from plants [5]. Numerous plant extracts have been widely reported to have antidiabetic actions and they have the potential to be an excellent therapeutic agent for diabetes mellitus with lesser side effects [6]. Therefore, surveillance for a new antidiabetic agent derived from plant sources is extremely urgent.

*Lasia spinosa* is a perennial herb of Tropical Asia and South-East Asia. Its an edible plant known as Kattosh or Kantakachu of the Araceae family. Different parts of this plant are traditionally used in various diseases [7]. Some Bangladeshi tribe people use its corm, stem, and rhizome in throat infection piles, cough, itching, rubella, measles, and skin lesions [8]. The leaves and tubers are also used in constipation, rheumatoid arthritis, and bacterial and fungal infections [9,10,11]. Furthermore, the phytochemical status, nutritional values, and antioxidant and anticancer properties of *L*. *spinosa* are also evaluated [12,13,14]. The antihyperglycemic effect of *L*. *spinosa* leaves has been reported by Hasan et al. [12] and Men et al. [15] investigated its antioxidative effects using n-hexane, ethyl acetate, and ethanol extract. It is recently reported the effect of *L. spinosa* stem ethanolic extract on STZ-induced pancreas and kidney lesions using a chemicobiological model [16]. As part of our continuous search, we have further extended our research how the *L*. *spinosa* methanol stem extract can help ameliorate integrated diabetes and diabetes-linked complications using both in vitro and in vivo models.

## 2. Results

### 2.1. Phytochemical and Antioxidative Status

Appendix A provides an overview of the qualitative phytochemical examination of secondary metabolites in the ME_X_LS. Whereas the ME_X_LS did not include any saponins, phlobatannins, or cardiac glycosides but contained alkaloids, flavonoids, steroids, tannins, carbohydrates, and proteins.

As indicated in Table 1 and Figure 1, the GC-MS analysis of ME_X_LS revealed the existence of substances with retention times ranging from 9.675 to 24.618 (min). The compounds were found to have the highest peak areas for methyl α-d-galactopyranoside (4616107), methyl α-d-glucopyranoside (4616107), 13-docosenamide, (Z)-(13399817), and 9-octadecenamide, (Z)-(13399817)**.**

Antioxidative capacities of ME_X_LS are summarized in Table 2. The total flavonoids content (TFC) in ME_X_LS was determined to be 277.50 ± 32.25 mg/g dry weight (rutin equivalent). ME_X_LS’s precise total phenolic content was determined to be 154.06 ± 0.62 mg/g dry weight (gallic acid equivalent). ME_X_LS was discovered to have a total antioxidative capacity (TAC) of 177.08 ± 3.20 mg/g dry weight (ascorbic acid equivalanet). According to the catechin standard curve (y = 4 × 10^−5^x + 0.0432, R^2^ = 0.9889), the total proanthocyanidine content (TPACC) of ME_X_LS was calculated, and it was discovered that ME_X_LS had the TPACC amount of 337.50 ± 29.92 mg/g of dry weight in comparison to catechin.

DPPH free radical scavenging effect, nitric oxide scavenging effect, iron chelating effect, and hydroxyl radical scavenging effects of ME_X_LS are summarized in Table 2. For ME_X_LS and ascorbic acid, the standard antioxidative agent, the half maximum inhibitory concentration (IC50) to scavenge the DPPH radicals was determined to be 14.38 ± 2.18 μg/mL and 9.22 ± 0.80 μg/mL, respectively. The IC50 value for NO scavenging assay was found to be 14.4 ± 0.17 μg/mL which was close to that of quercetin (3.06 ± 0.64 μg/mL). The reference antioxidant agent ascorbic acid had an IC50 value for iron chelation of 48.39 ± 1.87 μg/mL compared to ME_X_LS’s IC50 value of 60.61 ± 2.31 μg/mL. ME_X_LS had an IC50 value of 184.40 ± 0.71 μg/mL. For both ME_X_LS and catechin, the hydroxyl radicals were seen to be reduced. The IC50s for ME_X_LS and catechin were 184.40 ± 0.71 μg/mL and 163.87 ± 3.35 μg/mL, respectively.

Table 2 also provides an overview of ME_X_LS’s impact on the suppression of protein denaturation, lipid peroxidation, and membrane stability. In terms of IC50 values, the membrane stabilizing activities of ME_X_LS, and the reference standard (ascorbic acid) were, respectively, 9.73 ± 1.00 μg/mL and 1.73 ± 0.19 μg/mL. The correlation between the results is statistically significant (*p* < 0.05). In lipid peroxidation inhibition assay, ME_X_LS and standard catechin were found to show the IC50 values 59.53 ± 8.37 μg/mL and 44.72 ± 6.64 μg/mL, respectively. The IC50 values for ME_X_LS and the reference standard, ascorbic acid, for their ability to suppress protein denaturation were 394.77 ± 8.11 μg/mL and 124.10 ± 4.58 μg/mL, respectively.

### 2.2. Impact of ME_X_LS on α-Amylase Inhibition

The α-amylase inhibitory effects of ME_X_LS and reference standard acarbose are presented in Figure 2. With an increase in sample concentration, the percentage of α-amylase inhibition rises. Comparing ME_X_LS’s α-amylase inhibition to that of the common amylase inhibitory enzyme, acarbose, was statistically significant (*p* < 0.001).

### 2.3. Effect of ME_X_LS on the In Vivo Assays

In animal study, ME_X_LS at the highest dose of 2000 mg/kg BW was not determined to be toxic. As a result, the LD_50_ of the ME_X_LS extract may exceed 2000 mg/kg (2 g/kg). Body weight-changes of the animal groups throughout the intervention period are shown in Figure 3. Except for the NC group, all groups’ body weights were declined significantly (*p* < 0.05) after the first week. Except for the DC group, which gradually lost weight, the body weights of all groups were considerably (*p* < 0.05) higher after the fourth week of intervention.

Blood glucose levels of the animals throughout the intervention period are shown in (Figure 4a). Diabetic control (DC) group had a significantly higher DC group which was significantly (*p* < 0.05) higher than all groups. At the 1st week of intervention, the blood glucose level of different groups showed a significant (*p* < 0.001) decrement in blood glucose. Some of them were almost the same as the NC group. After the 4th week of intervention, blood glucose levels of all groups were less than 16 mmol/L except DC group. Figure 4b shows the oral glucose tolerance level at the third week of the trial. The ME_X_LS200 group’s glucose tolerance ability was significantly (*p* < 0.001) higher than other groups and it was near to the RC group.

The variations in the weights of the pancreas, kidney, and liver are shown in Table 3. The pancreas weights of the ME_X_LS200 group were significantly (*p* < 0.001) higher than the DC group. The ME_X_LS50 and ME_X_LS100 groups’ kidney weights were considerably (*p* < 0.001) greater than the diabetic control group. ME_X_LS200 particularly restored kidney weights that were nearly identical to those of the NC and RC groups. However, liver weights of each group’s animals were higher than the DC group.

The effect of ME_X_LS on animal liver glycogen, serum alanine aminotransferase (ALT), aspartate aminotransferase (AST), creatinine, creatinine kinase-MB (CK-MB), lipid profile, lactate dehydrogenase (LDH), uric acid, and urea level is summarized in Table 4. The liver glycogen concentrations of ME_X_LS50, RC, and NC were significantly (*p* < 0.001) lower than DC group. Serum ALT levels of the treatment groups were significantly (*p* < 0.001) lower than the DC group. ME_X_LS50 was found to be the best among other doses for reducing ALT level. The treatment groups’ serum AST levels were considerably (*p* < 0.001) lower than those of the DC. ME_X_LS50 had the greatest effectiveness in controlling the AST levels of STZ-induced rats. Serum total cholesterol levels of the treatment groups were significantly (*p* < 0.001) lower compared to the DC group and ME_X_LS50 was found to be the most effective dose in reducing total cholesterol and triglycerides when compared with other groups. The HDL levels of treatment groups were displayed significantly (*p* < 0.001) higher than the DC group. Treatment groups’ LDL levels were found to be significantly lower than those of the DC group, and ME_X_LS50 had the lowest LDL levels than all other groups except the RC group. ME_X_LS was found to be effective in normalizing the LDH, CK-MB, creatinine, urea, and uric acid levels in which ME_X_LS100 showed the highest significant (*p* < 0.001) effect in comparison to the DC group. ME_X_LS50 has maximally improved the insulin homeostasis of STZ-induced diabetic rats and insulin level of ME_X_LS50 was significant (0.20 ± 0.04) compared to the DC (0.06 ± 0.02) group.

### 2.4. Effect of ME_X_LS on the Tissue’S Architectures

Figure 5 shows the architectural changes of the pancreas and scoring for the changes has been presented in Table 5. The size of the pancreatic islets of Langerhans was shown to be smaller in the DC group when it is compared to NC group. Degeneration was also discovered. In contrast, the other treatment groups were shown to exhibit less degeneration, and the area occupied by β-cells and the islet of Langerhans is much higher than in the DC group. Histopathological changes of kidney tissues and grading for the changes are presented in Figure 6 and Table 6, respectively. Data indicate that ME_X_LS50 was more promising than the two other doses in restoring kidney tissue damages, the higher tubular epithelial cell degeneration, tubular epithelial cell necrosis, and hyperemic interstitium vessels.

### 2.5. Effect of ME_X_LS on the Compounds-Proteins Interactions

The peroxisome proliferator-activated receptor gamma (PPAR, PDB ID: 3G9E), AMP-activated protein kinase (AMPK, PDB ID: 4CFH), and α-amylase enzyme was interacted with ten selected drugs to assess their antidiabetic potential (PDB ID: 1PPI). Five of them showed (Table 7) more binding affinity than the reference drug metformin with docking score, and the sample may have included a variety of bioactive metabolites that showed a remarkable effect as a treatment for diabetes. By the way, methyl α-d-galactopyranoside was thought to be a promising candidate based on its abundance and physiochemical characteristics; however, the compound showed the highest binding affinity to PPAR (Figure 7), AMPK (Figure 8), and 1PPI α-amylase (Figure 9), with docking scores of −5.218 Kcal/mol, −5.764 Kcal/mol, and −5.615 Kcal/mol, respectively. When all of them were combined, the compounds exerted synergistic effects. When all these factors were considered, the compounds acted on PPAR, AMPK, and α-amylase to provide an antidiabetic effect.

### 2.6. Effect ot ME_X_LS on the Pharmacokinetic Properties

The research has unveiled the drug-like characteristics of selected compounds using Lipinski’s rule of five and Veber’s rules (absorption, distribution, metabolism, and excretion/transport). Among them, only two compounds violated one rule, but the remaining compounds followed both Lipinski’s rule of five and Veber’s rule (Table 8). Hence, all compounds elucidated drug-like attributes and orally available.

## 3. Discussion

The extensive antidiabetic effects of ME_X_LS were investigated through its inhibitory action on α-amylase heading to an in vivo study in STZ-induced animal models which was verified by in silico tests. Most of the plants possess numerous secondary metabolites some of which are reported to show antidiabetic actions [17]. Flavonoids and their derivatives are known to have a wide range of biological and pharmacological activities such as antioxidant, anti-inflammatory, antimicrobial, and anti-diarrheal activities [18]. Oxidative stress is considered as a pivotal factor for several diseases including diabetes and thus the inhibition of intracellular free radical formation would provide a therapeutic strategy to prevent endothelial dysfunction in diabetes mellitus. Therefore, supplementation with antioxidants or development of antioxidant-coupled hypoglycemic agents from plant biometabolites is prospective for diabetic pharmacotherapy [19].

Oxidative stress in diabetes induces lipid peroxidation, and protein oxidation [20]. Lipid peroxidation has a high significance in the toxicology associated with redox imbalance [21] leading to stress-sensitive intracellular signaling pathway which plays a key role in the late complications of DM [22]. Inflammation, one of the inevitable consequences in DM, releases the lysosomal hydrolytic enzymes which destabilize the lysosomal membranes [23]. Therefore, membrane stabilizers are apparently signified to inhibit protein denaturation [24] and antidiabetic drugs such as TZDs, DPP-4 inhibitors, GLP-1 RAs, and insulin have proven for their anti-inflammatory properties [25]. The effect of ME_X_LS on lipid peroxidation inhibition, membrane stabilization, and protein denaturation inhibition might help ameliorate diabetic complications.

Several native medicinal herbs have a strong potential for suppressing the activity of α-amylase [26,27,28] which slows carbohydrate digestion and extends the overall carbohydrate digestion time resulting in a lower rate of glucose absorption [29]. The results of the study support that the α-amylase inhibitory action of ME_X_LS could be achieved through the decrease or inhibition of carbohydrates decomposition.

During the experimental period, it was observed that the body weight of the diabetic control (DC) group was drastically decreased. It is reported by researchers that weight loss in STZ-induced hyperglycemia is caused due to the increased muscle wasting and protein loss of the tissues. In diabetes mellitus, a reduction in body weight and a rise in food and water consumption are usually observed due to the destruction of β-cells, which may be due to metabolic changes triggered by a lack or deficiency of insulin [30]. As a result of a lack of carbohydrates for energy metabolism, structural protein breakdown may be the reason of the reduced body weight as seen in diabetic rats.

Streptozotocin (STZ)-induced diabetes mellitus caused by insulin deficiency leads to an increased blood glucose. In this investigation, it was revealed that ME_X_LS had a tremendous effect on the weekly blood glucose levels. The oral glucose tolerance test (OGTT) data indicated that the blood glucose was markedly decreased after treatment with ME_X_LS, suggesting that it can improve the sensitivity or stimulate secretion of insulin in the STZ-induced diabetic rats, because plasma glucose and insulin responses during this test reflect the ability of pancreatic β-cells to secrete insulin and the sensitivity of tissues to insulin [31]. All the groups had better glucose tolerance ability than the DC group. This could be the recovery of the pancreas and secretion of insulin into the blood due to the administration of ME_X_LS to the STZ induced diabetic rats.

The kidney weight of the treatment groups was close to that of NC group and significantly lower than DC group indicating the kidney enlargement of DC group. A possible mechanism for renal enlargement could be the direct effect of growth hormone (GF) and insulin-like growth factor (IGF)-1 [32]. The decrease in the weight of the pancreas of the DC group could be attributed to the disruption and disappearance of pancreatic islets and selective destruction of insulin-producing cells [33]. A mixed liver weight is recorded with the treatment of ME_X_LS in the experiment. Liver weight can be decreased due to the increased triglyceride accumulation. In this study, an enlarged liver is noted which may be due to the increased influx of fatty acids into the liver induced by hypoinsulinemic and the low capacity of excretion of lipoprotein secretion from the liver [34]. Additionally, a large amount of glycogen was reserved in the STZ-induced DC group when they fasted 24 h before sacrifice. The high liver glycogen level in diabetic rats may be attained either by an increase in gluconeogenesis or hyperglycemia [35]. Therefore, the low level of liver glycogen of the treatment groups designates the antidiabetic potentiality of ME_X_LS.

An increase in the activities of plasma ALT and AST is usually approached as liver dysfunction caused by the liver necrosis in STZ-induced diabetic rats [36]. It is also suggested that the elevated levels of AST and ALT in the serum of STZ-induced diabetic rats are escalated by hepatocellular damage. Restoration of elevated ALT and AST levels in the treatment groups of our experiments might be consistent with the improvement of insulin-resistant condition which is supported by the significant increase in insulin secretion by ME_X_LS50 and ME_X_LS100 because insulin is the pivotal factor considered for the improvement of diabetic condition.

The risk of cardiovascular complications is two to two-and-a-half times greater in people with T2DM compared with the nondiabetic population [37]. Increased serum creatinine kinase (CK-MB) and lactate dehydrogenase (LDH) levels may serve as a marker for cardiovascular risk and cardiac muscular damage [38]. In this investigation, serum LDH and CK-MB activities were found to be increased in STZ-induced diabetic rats, possibly due to myocardial dysfunction because it has been previously reported that serum LDH and CK-MB activities were found to be increased in cardiomyopathy [31].

Serum lipids are known to be elevated during severe diabetes and have been implicated in the development of atherosclerosis [39]. The serum lipid levels (cholesterol, TG, and LDL) of the NC, RC, and extract-treated diabetic rats were significantly reduced after four weeks of treatment which is converse to the DC group. Diabetes-induced hyperlipidemia is caused by an excess mobilization of fat from the adipose tissue because of glucose underutilization. It was also reported that diabetic patients, especially those with very poor glycemic control, may have increased LDL that is reduced by treatment of their diabetes, and this is due to effects on either the LDL or the receptor [40]. On the other hand, HDL levels were decreased in the DC group when compared with other groups and several investigations suggest that HDL levels decrease in diabetic patients [40].

This investigation revealed that induction of diabetes resulted in the elevation of serum urea, uric acid, and creatinine concentrations. Abnormalities of these parameters are considered as significant markers of renal dysfunction and various studies confirm that raised plasma creatinine and urea levels in diabetic patients may indicate a prerenal problem [41]. ME_X_LS administration resulted in a decrease in these parameters, a finding that agreed with that of the renal protective activity of ME_X_LS.

Histopathology of the pancreas in control animals showed normal pancreatic parenchyma cells and islet cells. In diabetic control, the pancreas section showed islet cell necrosis, edema, inflammation, and changes in exocrine and endocrine components. The groups that were treated with ME_X_LS showed less islet cell necrosis. Edema, inflammation, and changes in exocrine and endocrine components were less common in the treatment groups. The RC group showed the high recovery of damaged cells and ME_X_LS100, ME_X_LS200 showed significant recovery. Other groups showed moderate recovery than the DC group. Histological study of the normal kidney revealed normal kidney cells without any inflammatory changes. Kidneys of untreated diabetic rats showed degenerated glomeruli infiltrated by inflammatory cells, and other changes were edema, cell necrosis and degeneration of epithelia. The groups that were treated with ME_X_LS showed features of healing that is normal glomerulus, less inflammatory cells and recovery of edema, cell necrosis and degeneration of epithelia was common. ME_X_LS200 group highly showed recovery and other groups showed moderate recovery. ME_X_LS has favorable effect to inhibit the histopathological changes of the pancreas and kidney in STZ-induced diabetes. Antidiabetic action of *L. spinosa* in diabetic rats may be possible through the insulinomimetic action or by other mechanism such as stimulation of glucose uptake by peripheral tissue, inhibition of endogenous glucose production, or activation of gluconeogenesis in liver and muscle [42].

Molecular docking study plays a pivotal role in structural molecular biology and computer-assisted drug design (CADD) to develop a new drug, while the molecular docking tool evaluates the prediction of binding interactions of new compounds against relevant proteins [43]. Moreover, the possible molecular mechanism of action of different pharmacological activities is determined comprehensively through molecular docking [44]. However, to get the association with the results of the study, molecular docking was conducted to comprehend the molecular mechanism better. To reach a better perception of biological activity, a total of 11 compounds were selected by PASS prediction from GC-MS data for ME_X_LS. These compounds were examined against the active site of peroxisome proliferator-activated receptor gamma (PPARγ, PDB ID 3G9E), AMP-activated protein kinase (AMPK, PDB ID: 4CFH), and α-amylase enzyme (PDB ID: 1PPI). Some compounds exhibited higher binding affinity than the standard drug metformin. Peroxisome proliferator-activated receptor (PPARγ) activates some genes in tissues that result in an increase in glucose and lipid uptake, decreases free fatty acid concentration, and subsequently decreases insulin resistance [45]. PPARγ is a target for insulin sensitizing drugs such as glitazones, which improve plasma glucose maintenance in patients with diabetes. Synthetic ligands have been designed to mimic endogenous ligand binding to a canonical ligand-binding pocket to hyperactivate PPARγ [46]. AMPK, a heterotrimer, consisting of a catalytic α-subunit and regulatory β- and γ-subunits, is an energy-sensing enzyme that is activated when cellular energy levels are low, and it signals to stimulate glucose uptake in skeletal muscles, fatty acid oxidation in adipose (and other) tissues, and reduces hepatic glucose production. AMPK is an evolutionarily conserved serine/threonine kinase whose activation elicits insulin-sensitizing effects, making it an ideal therapeutic target for T2D [47]. Metformin was shown to activate the AMP-activated protein kinase (AMPK) in intact cells and in vivo, a drug widely used to treat type 2 diabetes. To treat type 2 diabetes, human pancreatic α-amylase (HPA) in the small intestine inhibition is important because pancreatic amylase correlates with an increase in postprandial glucose levels. In this investigation, we observed a decrease in the blood glucose level in diabetic animals when treated with the title compounds. The possible mechanism by which the title compounds bring about their hypoglycemic action may be by binding to PPAR-γ receptor or AMPK receptors or 1PPI receptor as a potent agonist or antagonist and increasing the body’s insulin sensitivity [48].

The pharmacokinetic properties of different phytocompounds of ME_X_LS might be depicted according to Lipinski’s rule of five, orally administered drugs should have a molecular weight ≤ 500 amu, hydrogen bond acceptor site ≤ 10, hydrogen bond donor sites ≤ 5, and lipophilicity value, Log P ≤ 4.15 and Veber’s rule of two (number of rotatable bonds ≤ 10, topological polar surface area ≤ 140). If any drugs/compounds violate all of these rules, they will not be considered as good oral bioavailability fives [49,50]. This study demonstrated that some compounds in ME_X_LS indicate good oral bioavailability from the studied bioactive compounds. Therefore, these phytocompounds could be considered for promising drug candidates with good oral bioavailability, based on a study illustrated as a drug-like characteristic. Experimental evidence obtained from this study demonstrate that ME_X_LS possesses the potential to attenuate the complications of diabetes and diabetes-related diseases which is affirmed by the histopathological examination, and computational studies.

## 4. Materials and Methods

### 4.1. Chemicals and Reagents

The chemicals and reagents were ensured as analytical grade until unless specified individually.

Iodine, starch (ACS reagents, soluble starch, Catalog No. 1012520100), methanol, α-amylase (Catalog No. 86250, powder form, originated from *Aspergillus oryzae*), Potassium dihydrogen phosphate (KH_2_PO_4_), potassium monohydrogen phosphate (KHPO_4_), acarbose (Catalog No. A8980), tris-HCL buffer, Tri chloroacetic acid (TCA), butylated hydroxytoluene (BHT), potassium chloride (KCl), ferric chloride (FeCl_3_), hydrochloric acid (HCl), thiobarbituric acid (TBA), bovine serum albumin (BSA, lyophilized powder, Catalog No. A9418), aspirin, sodium chloride (NaCl), sodium-citrate dihydrate (99%), sodium phosphate (Na_3_PO_4_) buffer, and dextrose buffering phosphate, glucose, xylene, wax/paraffin, glycerin, picric acid, streptozotocin (STZ), formaline (100%), Anthrone, and Mayer’s affixative hematoxylin, Eosin, and DPX were procured from Sigma-Aldrich (Sigma-Aldrich, Inc., St. Louis, MO, USA). Metformin hydrochloride was kindly donated by Square Pharmaceuticals Ltd., Dhaka, Bangladesh. Food grade sugar and D-fructose were procured from local suppliers.

### 4.2. Collection and Extraction of Plant Material

*Lasia spinosa* (L.) Thwaites (local name Kattosh) stems were collected from Chittagong district (Fatehabad region), Bangladesh. The plant was identified by Dr. Sheikh Bokhtear Uddin, Taxonomist, and Professor, Department of Botany, University of Chittagong. A sample specimen of the plant was preserved with an accession number LAMLS-A120 for future reference. After washing and cleaning, the plant stem was chopped and dried at room temperature under shade and in an oven at 40 °C overnight. The dried samples were ground by using an electric grinder (BI-DTool pulvirizer, Xinxiang Dongzhen Machinery Co., Ltd., Xinxiang, China) to prepare crude extract following the protocol of Altemimi et al., [51]. The resultant powder (1200 g) was immersed for 7 days with intervals of 2 to 3 days in 2.0 L of methanol at room temperature (23 ± 0.5 °C) with periodic stirring. A rotary evaporator (RE200, Bibby Sterling, UK) was used to concentrate the collected supernatant while it was under reduced pressure and below 50 °C. The ME_X_LS concentrated crude extract was stored in a Petri dish and allowed to dry at 37 °C to allow all solvents to completely evaporate. In preparation for future use, 12.0 (yield 1.0%) g of concentrated semisolid dark-brown methanolic stem extract (ME_X_LS) were kept at 4 °C.

### 4.3. Assay for Phytochemical Groups Status

To carry out the phytochemical status of the crude MExLS, screening tests were accomplished through the established methods [52,53,54]. In each test, 10% (*w*/*v*) solution of the extract was taken [55] and tested for alkaloid, flavonoids, steroids, tannins, saponins, phlobatanins, glycosides, carbohydrates, and proteins.

### 4.4. Gas Chromatography-Mass Spectroscopy (GC-MS) Analysis

The biometabolites extracted from the leaves of *L. spinosa* methanol stem (ME_X_LS) were analyzed by gas chromatography (GC-2010 plus, Shimadzu Corporation, Kyoto, Japan), coupled with a mass spectrometer (GCMS-TQ 8040, Shimadzu Corporation, Kyoto, Japan). A fused silica capillary column (Rxi-5ms; 30 m, 0.25 mm ID, and 0.25 μM) was used for GC, maintaining sample inlet temperature at 250 °C. A 1.0 μL sample was injected in split less mode. The oven temperature was programmed as 75 °C (1 min); 25 °C, 125 °C (1 min); 10 °C, 300 °C (15 min). The aux (GC to MS interface) temperature was set to 250 °C. Total run time was 36.50 min and column flow rate 1.5 mL/min He gas. An electron ionization (EI) type mass spectroscopy (MS) was used in Q3 scan mode. Moreover, 200 °C ion source temperature, 250 °C interface temperature, 1.17 kV detector voltage, and 50–1000 *m*/*z* mass range were set for MS. The individual compound with *m*/*z* ratio was searched in “NIST-MS Library 2014. Total ionic chromatogram (TIC) was used to determine the peak area as well as the percentage amounts of each compound.

### 4.5. Determination of the Antioxidative Effects of ME_X_LS

#### 4.5.1. Evaluation of Total Phenolic Content (TPC), Total Flavonoid (TF), Total Antioxidative Capacity (TAC) and Total Proanthocyanidin Content (TPACC)

Total phenolic content of ME_X_LS was determined according to a method established by Kaur and Kapoor (2002) [56]. A total of 2.5 mL of Follin-Ciocalteau reagent was applied to each test tube together with 5 mL of ME_X_LS (8 mg/10 mL) or the standard (gallic acid) to determine the total antioxidant capacity. Then, 2.5 mL of Na_2_CO_3_ solution at 7.5% was added to each test tube. All the test tubes were incubated for 20 min at 25 °C and read at 760 nm using spectrophotometer (UV-VIS 1280 spectrophotometer, Shimadzu Corporation, Japan) to measure the TPC using the following equation: C = (c × V)/m
where C = TPC (mg/g ME_X_LS in GAE), c = concentration of sample obtained from calibration curve (mg/mL), V = volume of the sample, m = sample weight (g). The total phenolic content of ME_X_LS was expressed as gallic acid equivalents (GAE) per gram of ME_X_LS.

The established method by Kumaran and Karunakaran was modified to determine the total flavonoid content of ME_X_LS [57]. Briefly, test tubes were filled with either 1 mL of standard solution or 1 mL of ME_X_LS (8 mg/10 mL) and added 3 mL of methanol followed by adding 200 μL of 10% AlCl_3_ and 200 μL of 1 M CH_3_COOK. Then, 5.6 mL of distilled water was added to the tubes and incubated for 30 min to read at 415 nm. Rutin was used as standard for calculation as follows: C = (c × V)/m
where C = TFC (mg/g ME_X_LS in Rutin), c = concentration of sample obtained from calibration curve (mg/mL), V = volume of the sample, m = sample weight (g). The results were expressed as mg of Rutin equivalent (RE)/g of dried MEXLS.

The total antioxidant capacity of ME_X_LS was measured by the phosphomolybdenum method [58]. At first, 0.5 mL of ME_X_LS or standard was added to 3 mL of the reaction mixture containing 1 mL of 1.8 M H_2_SO_4_, 1 mL of 0.084 M of dipotassium hydrogen phosphate, 1 mL of 3% ammonium molybdate. After that, the test tubes were incubated at 95 °C for 10 min and cooled (at room temperature 25 ± 1 °C) for 10 min to read at 695 nm. Ascorbic acid was used as standard in the calculation as follows: C = (c × V)/m
where C = TAC (mg/g ME_X_LS in ascorbic acid), c = concentration of sample obtained from calibration curve (mg/mL), V = volume of the sample, m = sample weight. The results were expressed as mg of ascorbic acid equivalent per gram of dried ME_X_LS.

Proanthocyanidins were determined by the method of Broadhurst [59]. Briefly, 300 μL of ME_X_LS or standard was added to 1.8 mL of 4% *w*/*v* vanillin—methanol solution in test tubes. Then, 900 μL of 4% *v*/*v* hydrochloric acid—methanol solution was mixed to the reaction mixture and incubated for 15 min at room temperature to read at 500 nm. Catechin was as standard for TPACC calculation using the following equation: C = (c × V)/m
where C = total pro-anthocyanidin content (mg/g ME_X_LS in catechin), c = concentration of sample obtained from calibration curve (mg/mL), V = volume of the sample, m = sample weight. The results were expressed as mg of catechin equivalent per gram of dried ME_X_LS.

#### 4.5.2. Assay of DPPH Free Radical Scavenging Effect of ME_X_LS

The free radical scavenging activity of, as shown in Table 1, 1-diphenyl-2-picrylhydrazyl (DPPH) of ME_X_LS was determined by Blois’s [60] protocol. The hydrogen atom donating ability of the ME_X_LS was determined by the decolorization of 2,2-diphenyl-1-picrylhydrazyl (DPPH) methanol solution. DPPH produces violet/purple color in methanol solution and transforms to shades of yellow color in the presence of antioxidants. A solution of 0.1 mM DPPH in methanol was prepared, and 2.4 mL of this solution was mixed with 1.6 mL of MExLS in methanol at different concentrations (50–800 μg/mL). The reaction mixture was vortexed thoroughly and left in the dark at RT for 30 min to read at 517 nm. The percentage of DPPH free radical scavenging activity was calculated using the following formula:% of DPPH free radical scavenging activity = {(A_0_ − A_1_/A_0_} × 100
where A_0_ is the absorbance of the control, and A_1_ is the absorbance of the ME_X_LS/standard. Then % of inhibition was plotted against concentration, and from the graph IC_50_ was calculated.

#### 4.5.3. Assay of Iron Chelating Activity of ME_X_LS

The iron-chelating effect of ME_X_LS was evaluated by Benzie and Strain’s method [61]. Briefly, 2 mL of ME_X_LS or standard/ascorbic acid (50–800 μg/mL) was mixed with 1 mL of methanol dissolved in 1,10-phenanthroline. After that, 2 mL of 200 μM FeCl_3_ solution was added to all test tubes and the reaction mixture was incubated at 25 °C for 10 min to read at 510 nm. Control was prepared in the same way excluding sample or standard. The iron-chelating activity was calculated through the following equation,
Percentage of iron chelating activity = {(A_1_ − A_0_)/A_0_} × 100,
where A_1_ = Absorbance of ME_X_LS, and A_1_ = Absorbance of control.

#### 4.5.4. Assay of Nitric Oxide Scavenging Activity of ME_X_LS

The nitric oxide scavenging activity of ME_X_LS was estimated by Marcocci’s method [62]. Briefly, 1 mL of ME_X_LS/standard solution (18.25–300 μg/mL) was added with 1 mL of 5 mM sodium nitroprusside (20 mM phosphate buffer, pH 7.4) solution and incubated at 25 °C for 150 min. After the incubation period, 2 mL of Griess reagent was added to all the test tubes and read at 546 nm. Quercetin was used as standard to calculate the iron-chelating effect using the formula:Nitric oxide scavenging activity = {(A_0_ − A_1_)/A_0_} × 100
where A_0_ = absorbance of control and A_1_ = absorbance of the extract/standard.

#### 4.5.5. Assay of Hydroxyl Radical Scavenging Activity of ME_X_LS

The crude ME_X_LS extract (1 mg/mL),100 μL was mixed with 500 μL of 2-deoxyribose (2.8 mM) in phosphate buffer (50 mM, pH 7.4), premixed ferric chloride (200 μL) and EDTA solution (1:1 *v*/*v*) and 100 μL H_2_O_2_ solution (200 mM). To trigger the reaction, 100 μL ascorbate (300 mM) was added and incubated for 1 h at 37 °C. From this mixture 0.5 mL was taken out and TCA (1 mL) and TBA solution (1 mL) were added to the reaction mixture. The mixture was heated in a water bath for 15 min and the absorbance was measured at 532 nm. The blank was prepared with 100 μL of plant extract with methanol instead of Fenton reaction mixture [63]. Hydroxyl radical scavenging activity = {(A_0_ − A_1_)/A_0_) × 100}; where A_0_ is the absorbance of the control reaction, A_1_ is the absorbance in presence of the extract samples and the reference standard.

#### 4.5.6. Assay of Membrane Stabilization, Lipid Peroxidation, and Protein Denaturation Inhibition

The membrane-stabilizing activity was determined by the method of Sadique et al. and Oyedapo et al. [64,65]. Four milliliter of fresh blood collected from volunteers was mixed with 4 mL of Alsever solution. The solution was centrifuged at 3000× *g* for 10 min and the packed cells were washed with isosaline. After preparing a 10% *v*/*v* suspension solution, 0.5 mg/mL ME_X_LS/or standard was dissolved in 10 mL distilled water to give a final concentration of 500 μg/mL as the stock solution. Serial dilution was carried out to obtain the concentrations 250–31.25 mg/mL. Sample/standard and respective solution (1 mL) were placed into test tubes with 0.5 mL RBC suspension. Then 2 mL of hypotonic solution was dissolved in phosphate buffer (10 Mm) and the reaction mixture was incubated for 30 min at 37 °C. After incubation, the solutions were centrifuged at 3000× *g* for 20 min and the supernatant was collected to read at 560 nm. Membrane stabilization activity was calculated using the following formula:Membrane stabilization activity = {(A_0_ − A_1_/A_0_)} × 100
where A_0_ is the absorbance of control and A_1_ is the absorbance of sample.

Lipid peroxidation inhibition assay of ME_X_LS was performed by the method established by [66]. First, 100 μL of ME_X_LS/standard was mixed with 500 μL of Bovine brain homogenate solution in a test tube to which 200 μL of 0.2 mM ferric chloride was added. Afterward, the reaction mixture was incubated at 37 °C for 30 min and added to 2 mL of 1% ice-cooled TBA-TCA-BHT solution and incubated at 90 °C for 60 min. Then the reaction mixtures were centrifuged at 3000 rpm for 10 min and supernatants were collected to read at 532 nm. Lipid peroxidation activity was calculated using the following formula: Lipid peroxidation activity = {(A_0_ − A_1_/A_0_)} × 100
where A_0_ and A_1_ are the absorbances of control and sample, respectively.

Protein denaturation inhibition assay was accomplished by Bovine serum albumin assay (BSA) reported by Williams et al. [67]. ME_X_LS (0.5 mg/mL) or standard (aspirin) was dissolved at 10 mL distilled water to give the final concentration 500 μg/mL as a stock solution and the serial dilution was carried out to obtain concentrations 250–31.25 mg/mL. The reaction mixture contained 0.45 mL BSA and 0.05 mL ME_X_LS or aspirin, taken in test tubes (triplicate) of different concentrations. Each solution was attuned to pH 6.3 by using 1 N HCl. The mixtures were incubated at 37 °C for 20 min and heated at 57 °C for 30 min. Then phosphate buffer (2.5 mL) was added, and the absorbance was measured at 660 nm. Protein denaturation inhibition was calculated using the following formula:Protein denaturation activity = {(A_0_ − A_1_/A_0_)} × 100
where A_0_ and A_1_ are the absorbances of control and sample, respectively.

#### 4.5.7. Assay of α-Amylase Inhibitory Effect of ME_X_LS

The α-Amylase inhibition activity was carried out using the starch-iodine method [68]. ME_X_LS, standard and respective solvent (1 mL) of different concentrations was placed into test tubes (triplicate) to which 20 μL of α-amylase solution (10 mg/mL) dissolved in phosphate buffer (0.02 M, pH 7.0 with sodium chloride 0.006 M) was added and incubated for 10 min at 37 °C. After incubation, 200 μL of 1% starch solution was added to each test tube and the mixtures were re-incubated for 1 h at 37 °C, and then 200 μL of 1% iodine solution was further added followed by adding 10 mL of distilled water to each test tube. Finally, the absorbance of the reaction mixtures was taken at 565 nm by using a spectrophotometer. Sample, substrate, and α-amylase blank were taken under the same conditions.

### 4.6. Animal Model Experiments

Male adult albino rats of the Wistar strain (avg. BW 150–200 g) were used for the study. They were collected from the animal breeding unit of Jahangirnagar University, Dhaka. The animals were individually housed in polycarbonated cages filled with wood husk at a temperature around (25 ± 2 °C) and humidity 55–60% in a 12 h light-dark cycle. All animals were fed with a commercial rat pellet diet during the entire experimental period. All targeted animal experimentations complied with the rules and regulations of the institutional Animal Ethics Review Board, University of Chittagong (approval no EACUBS2018-5 as of 19 September 2018).

#### 4.6.1. Assay for Acute Toxicity

According to the “Organization for Environmental Control Development” requirements, the acute toxicity study was carried out in a lab setting under standard settings (OECD: Guidelines 420; Fixed-Dose Method). Five rats from each of the allocated animals received a single oral dosage of ME_X_LS (500–2000 mg/kg BW). Before administration of the ME_X_LS, rats were kept fasting overnight, and food was also delayed for between 3 and 4 h. After administration, food was withheld for a further 3–4 h. Experimental animals were observed individually during the first 30 min after dosing, periodically for the first 24 h, with monitoring for possible unusual responses including changes in eyes, skin and fur, autonomic and central nervous systems, respiratory system, allergic syndromes and behavior patterns, and mortality over the next 72 h. The median lethal dose (LD50 > 2.0 g/kg) was used as an effective therapeutic dose [69].

#### 4.6.2. Induction of Diabetes in Animal Model

After 1 week of fructose feeding, overnight fasted male Wistar albino rats were induced diabetes with an intraperitoneal injection of single dose streptozotocin (STZ) at 50 mg/kg body weight [70]. Where STZ was dissolved in 0.1 M citrate buffer with pH 4.5. Only citrate buffer was injected into the normal control group at the same volume as STZ [71]. All the animals were provided with free access to water and pellet diet after 30 min of administration of STZ. The animals were kept under strict observation, and seven days after STZ injection, tail prick method was used to measure the fasting blood glucose level. Animals with fasting blood glucose levels (tail-prick method, Glucoplus Inc., Saint-Laurent, QC, Canada) of >300 mg/dL were considered as diabetic and selected for the intervention.

#### 4.6.3. Animal Grouping, Intervention, and Oral Glucose Tolerance

After measuring fasting blood glucose, the animals were divided into six groups (*n* = 6) where normal control group was only nondiabetic. Normal control group (NC) was treated with only citric buffer; diabetic control group (DC) received no treatment, reference control group (RC) received metformin at 125 mg/kg BW, sample treated groups were ME_X_LS50, ME_X_LS100, ME_X_LS200 which respectively received 50 mg/kg BW, 100 mg/kg BW and 200 mg/kg BW of ME_X_LS. In a 4 weeks intervention, fasting blood glucose, body weights, foods, and fluid intakes were recorded every week.

The oral glucose tolerance was tested at the end of the third week. Twelve-hour fasted animals were administered a single dose of glucose (2 g/kg BW) and blood samples were collected from the tail-pricks at 0, 30, 60, 90, and 120 min after glucose administration to record blood glucose levels [72].

#### 4.6.4. Sacrifice of Animals, Collection of Blood, Tissues, and Organs for Analyses

Animals were sacrificed using halothane anesthesia (1% halothane) at the end of the intervention and blood, pancreas, liver, and kidney were collected. Blood was collected through cardiac puncture and centrifuged at 3000 rpm for 15 min at 20 °C to separate the serum which was preserved at −20 °C for further analysis. The collected livers were washed with 0.9% NaCl, wiped out, weighed, and preserved at −40 °C for liver glycogen estimation. The pancreas and kidney were weighed and preserved in 10% buffered formalin for histopathological analysis [73]. Serum alanin aminotransferase (ALT), aspartate aminotransferase (AST), creatine kinase-MB (CK-MB), serum creatinine (SC), high-density lipoprotein (HDL), low-density lipoprotein (LDL), lactate dehydrogenase (LDH), total cholesterol (TC), triglyceride (TG), serum urea (SU), serum uric acid (SUA), and serum insulin were measured by semi-autoanalyzer using reaction kits (Humalyzer 3000, Human, Wiesbaden, Germany). Liver glycogen concentrations were measured by a phenol sulfuric acid method as described by Lo et al. [74]. Briefly, the liver samples (about 80 mg) were transferred to test tubes containing 30% KOH (*w*/*v*), which was saturated with Na_2_SO_4_. Liver tissues were then boiled for 30 min until complete homogenization occurs. The homogenized mixture was cooled in ice. Glycogen was precipitated by adding 2 mL 95% ethanol and then incubated on ice for 30 min followed by centrifugation at 840× *g* (3000 rpm) for 20–30 min. The supernatants were discarded, and the precipitate was dissolved in 3 mL distilled water and the aliquot was then diluted to 1:4 (100 μL aliquot in 400 μL distilled water). Standard was also started from here. Both the sample and standards were run duplicate. One milliliter of 5% phenol and 5 mL 96–98% H_2_SO_4_ were added to the solution and allowed to stand for 10 min. The solutions were incubated at 25–30 °C for 15 min. Absorbance of the solutions was measured at 490 nm. Glycogen concentrations of tissues were calculated using the following equation:Liver glycogen (mg/g tissue) = A/k × V/v × 10^−4^/w
where k = slope of the standard curve, V = total volume (mL) of glycogen solution, v = volume (mL) of aliquot to which phenol-sulfuric acid solution is added, A = absorbance at 490 nm, w = sample weight (g).

#### 4.6.5. Assay for Tissue Architecture

To determine how ME_X_LS affected streptozotocin-induced diabetes, a pancreatic and kidney histopathological assay was conducted. The preserved tissues were cut into 3–5 M thick slices, fixed in paraffin wax, and dehydrated by ethanol and cleaned in xylene. These sections were then mounted on slides and stained for microscopic analysis to provide histopathology slides. Under an Olympus BX51 microscope, many cellular characteristics of the kidney and pancreas were examined, and an Olympus DP20 system was used to capture histopathology images.

### 4.7. Statistical Analysis

The data are displayed as a mean ± SD. One-way ANOVA and the Tukey’s multiple range post-hoc test were used to evaluate the data via statistical software (Statistical Package for Social Science, SPSS Version 22.0, IBM Corporation, Armonk, NY, USA). At *p* < 0.05, the values were deemed to be statistically different.

### 4.8. Computational Studies

#### 4.8.1. Molecular Docking Analysis

Based on a literature review to determine antidiabetic activity, the receptors and enzymes were chosen for molecular docking. The Protein Data Bank (PDB) online database (https://www.rcsb.org/ (accessed on 11 November 2021)) was used to import the crystal structures of the proteins PPAR (PDB ID 3G9E), AMPK (PDB ID: 4CFH), and -amylase enzyme (PDB ID: 1PPI) for the antidiabetic activity. The best binding sites were chosen using the online tool Pock Drug [75]. Additionally, the chemical structures of GC-MS characterized prevalent compounds of ME_X_LS were retrieved from the PubChem library (https://pubchem.ncbi.nlm.nih.gov/ (accessed on 11 November 2021). Molecular docking study was accomplished by using the modified protocol of Hossen et al., (2021). Briefly, the Schrodinger suites-Maestro 2017-1 was used for molecular docking. The ligand was prepared using the LigPrep tool, embedded in Schrodinger suite-Maestro v 11.1, where the following parameters were used for minimization: neutralized at pH 7.0 ± 2.0 using Epik 2.2 and the force field OPLS3 [76].

#### 4.8.2. Pharmacokinetic Properties Evaluation

The absorption, distribution, metabolism, excretion, and toxicity (ADME/T) properties of ME_X_LS were evaluated by Lipinski’s rule of fives and Veber’s rule (number of rotatable bonds; topological polar surface area). The ADME/T properties analysis was evaluated by SwissADME (http://www.swissadme.ch/ (accessed on 11 November 2021) [77].

## 5. Conclusions

A plant-derived drug seems highly attractive to treat type 2 diabetes due to the limited efficacy and high risk of adverse effects of synthetic antidiabetic drugs. This research has attempted to explore *L. spinosa* as a potential source of phytochemical possibilities to improve diabetes and diabetes-related complications. The investigation showed lower weekly blood glucose levels, decreased liver and cardiac markers, reduced lipid profile, and increased glucose tolerance ability in diabetic rats. Significant impact on restoring ALT to 60.60 (U/L), creatinine 0.55 (mg/dL), LDL 37.60 (mg/dL), and TC 54.80 (mg/dL) compared to the values 48.30(U/L), 0.60 (mg/dL), 38.8 (mg/dL), and 69.00 (mg/dL) by the reference standard implies a great potential of MExLS, especially the low dose in animal model. Most importantly, the histopathological morphology of pancreatic islets of Langerhans and kidney tissue has been improved among the animals treated with ME_X_LS. Moreover, from the in silico pass prediction, methyl-α-d-galactopyranoside has showed the highest binding affinity with all three receptor protein (docking score −5.218, −5.764, and −5.615 with PPARγ, AMPK and 1PPI receptors respectively) compounds from GC-MS are greatly impacted by antidiabetic activity. Further in vivo and computational studies of isolated pure compounds with a dose-response relationship mechanistics may lead to design a therapeutic agent from *Lasia spinosa***.**

## Figures and Tables

**Figure 1 pharmaceuticals-15-01466-f001:**
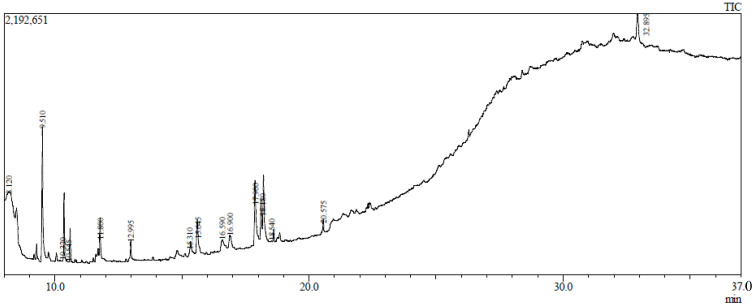
Gas chromatography-mass spectrometry profile of ME_X_LS was obtained from GC-MS with electron impact ionization (EI) method on a gas chromatograph (GC17A, Shimadzu Corporation, Kyoto, Japan) coupled to a mass spectrometer (GC-MS TQ 8040, Shimadzu Corporation, Kyoto, Japan).

**Figure 2 pharmaceuticals-15-01466-f002:**
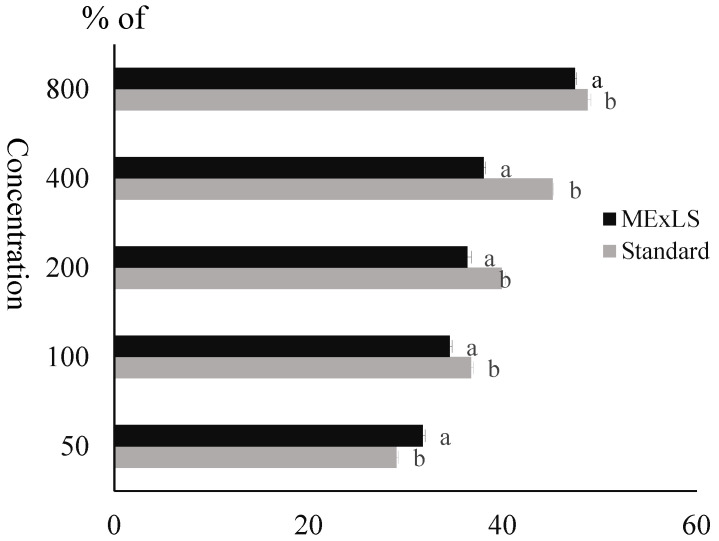
Percent of acarbose and ME_X_LS’s ability to inhibit α-amylase. A *t*-test was used to determine the significance of the data after being processed by the analytical program GraphPad Prism, where a significance threshold of *p* < 0.05 was used. The letters a and b over the bar graph indicate that the values are significant when they are compared to each other.

**Figure 3 pharmaceuticals-15-01466-f003:**
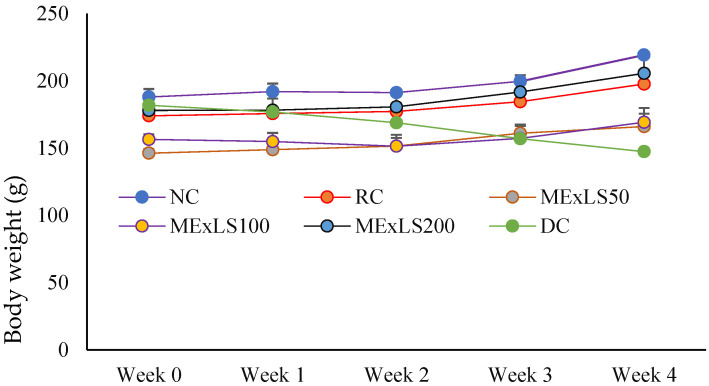
Changes of BW on intervention of ME_X_LS in Albino rats over four weeks (*n* = 5). Data are expressed as mean ± SEM. Data were analyzed by one way analysis of variance (ANOVA) using the statistical software SPSS followed by a Tukey’s Post Hoc test for significance. *p* < 0.05 was considered as significant.

**Figure 4 pharmaceuticals-15-01466-f004:**
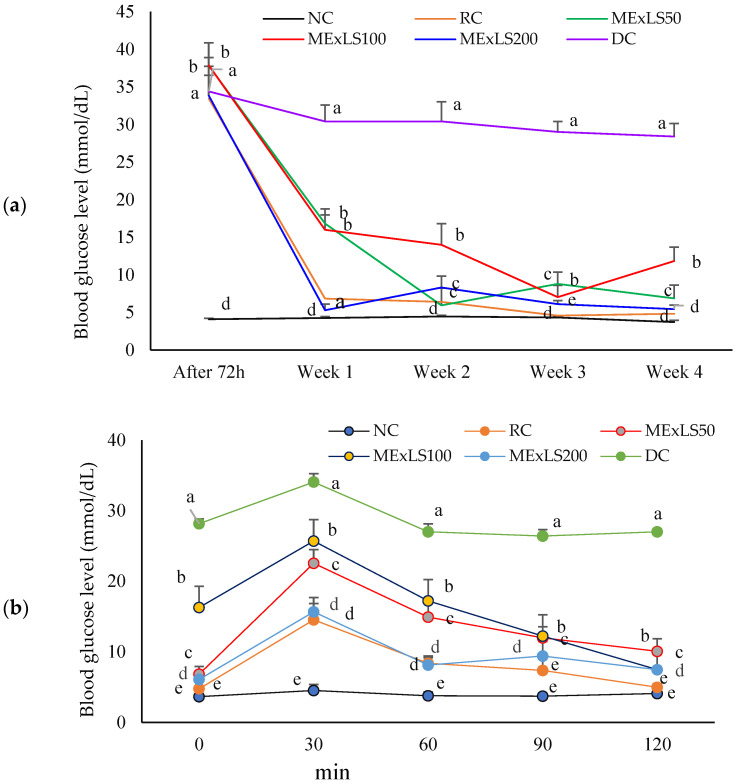
(**a**) Weekly blood glucose level for ME_X_LS intervention. (**b**) A four-week oral glucose tolerance test was conducted on albino rats under specific pressure and temperature conditions (*n* = 5). Mean ± SEM are used to express data. Data were analyzed by one way analysis of variance (ANOVA) using the statistical software SPSS followed by Tukey’s Post Hoc test. *p* < 0.05 was consideration as the level of significance. The superscript letter a-e on the lines indicate the significant differences between and among the groups.

**Figure 5 pharmaceuticals-15-01466-f005:**
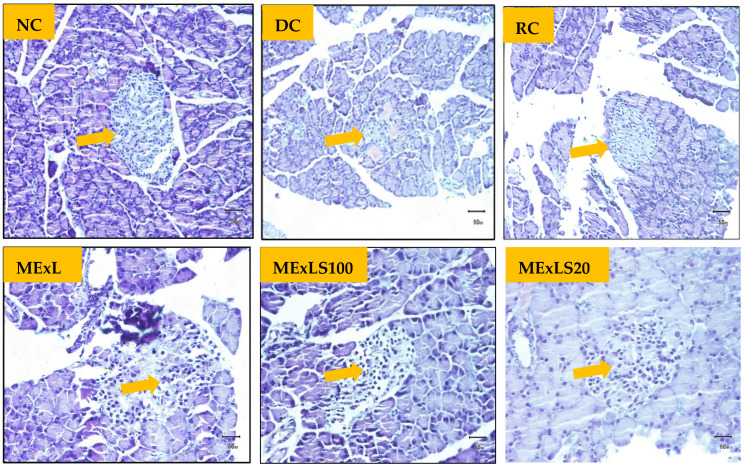
Pancreatic tissue from various experimental animals’ groups is shown in a histopathological image (H & E staining × 125) (microscopic resolution: 10 × 40). The pancreatic islet of Langerhans is depicted by an arrow. These pancreatic slices stained with PAS and counterstained with hematoxylin are exhibited under light microscopy.

**Figure 6 pharmaceuticals-15-01466-f006:**
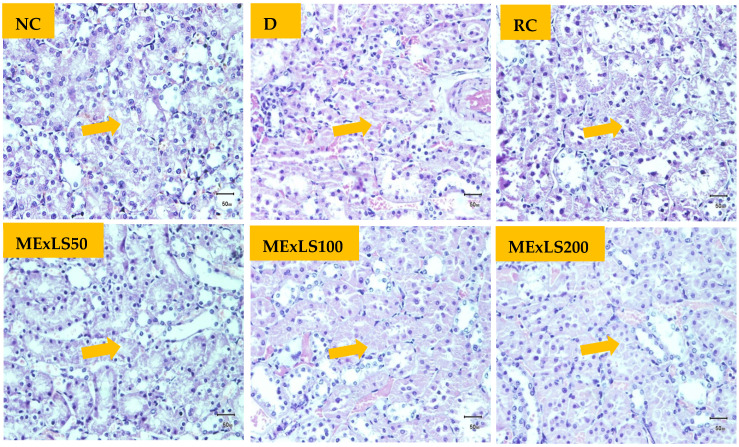
Image of the kidney tissues from the experimental animal groups as seen through histopathology. The glomerulus of a kidney cell is seen in the image (microscopic resolution: 10 × 40). Hematoxylin and eosin-stained rat kidney micrographs. The images displayed are the glomerulus slices counterstained with hematoxylin and stained with PAS. Arrow signs indicate the point of recovery from necrosis.

**Figure 7 pharmaceuticals-15-01466-f007:**
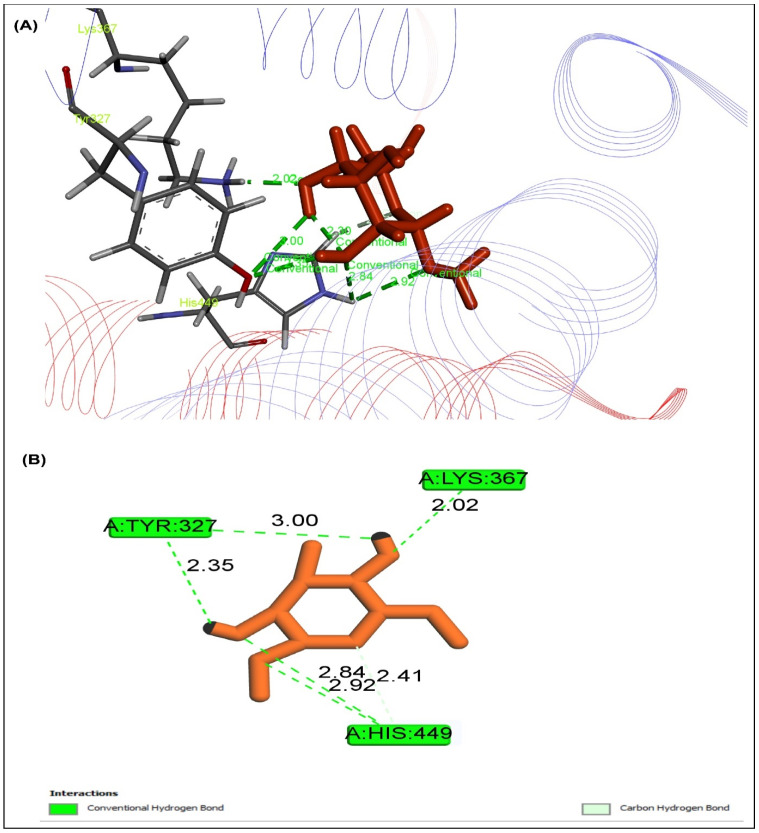
Docking study showed that the highest rank poses of methyl α-d-galactopyranoside docked with the active site PPARγ for antidiabetic potential in (**A**) 2D and (**B**) 3D molecular interactions.

**Figure 8 pharmaceuticals-15-01466-f008:**
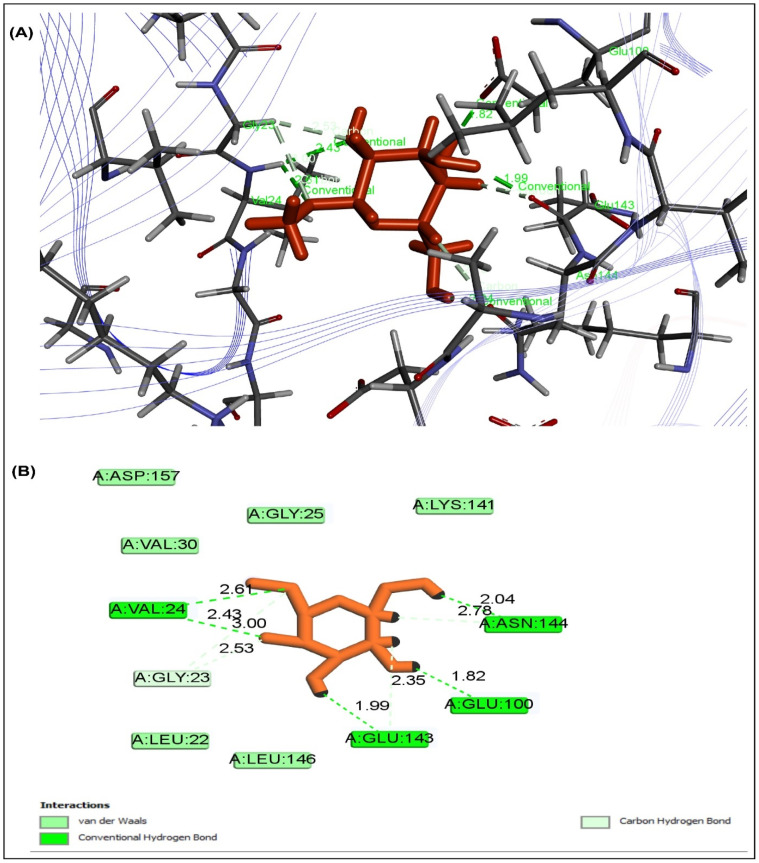
Docking study displayed that the highest rank poses of methyl α-d-galactopyranoside docked with the active site of AMPK for possible antidiabetic activity in (**A**) 2D and (**B**) 3D molecular interactions.

**Figure 9 pharmaceuticals-15-01466-f009:**
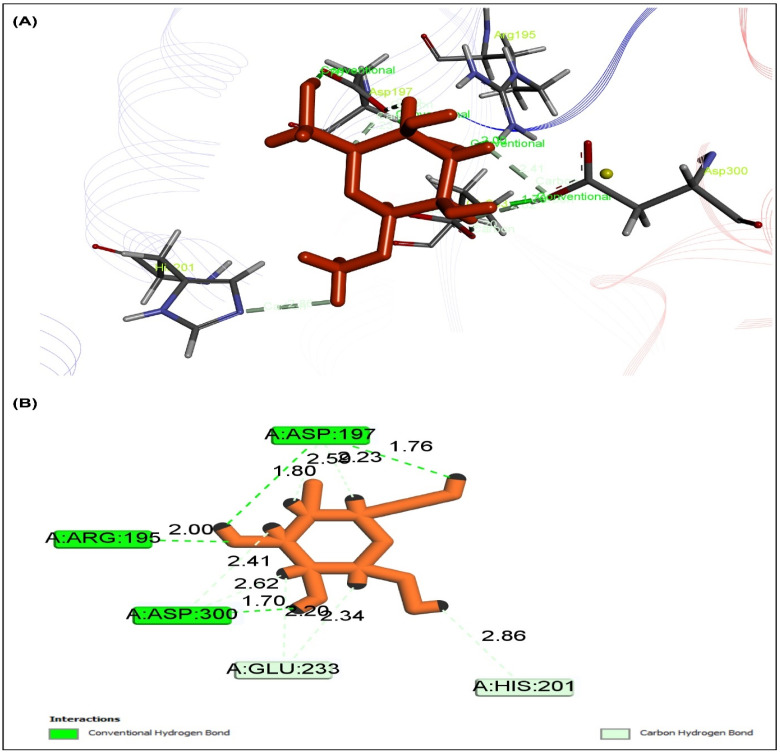
Docking study showed that the highest rank poses of methyl α-d-galactopyranoside docked with the active site 1PPI α-amylase for antidiabetic potential in (**A**) 2D and (**B**) 3D molecular interactions.

**Table 1 pharmaceuticals-15-01466-t001:** Compounds identified in the ME_X_LS by GC-MS.

S. N	Compounds Name	Molecular Formula	Molecular Weight (g/mol)	Running Time	Area
1	Methyl alpha-d-Galactopyranoside	C_7_H_14_O_6_	194.1825	9.675	4616107
2	Methyl alpha-d-Glucopyranoside	C_7_H_14_O_6_	194.1825	9.675	4616107
3	2-Penten-1-ol, (Z)-, TMS derivative	C_8_H_18_OSi	158.3134	10.140	74145
4	2-Butene-1,4-diol, TMS derivative	C_7_H_16_O_2_Si	160.29	10.140	74145
5	Silane, [[4-[1,2-*bis*[(trimethylsilyl)oxy]ethyl]-1,2-phenylene]*bis*(oxy)]bis[trimethyl-]	C_20_H_42_O_4_Si_4_	458.90	10.144	2431169
6	2-Buten-1-ol, (E), TBDMS derivative	C_10_H_22_OSi	186.3666	10.140	74145
7	2-[(2,4,4,6,6,8,8-Heptamethyl-1,3,5,7,2,4,6,8-tetroxatetrasilocan-2-yl)oxy]-2,4,4,6,6,8,8,10,10-nonamethyl-1,3,5,7,9,2,4,6,8,10-pentoxapentasilecane	C_16_H_48_O_10_Si_9_	653.316	12.169	30800
8	Phenethylamine, *N*-methyl-.β.,3,4-*tris*(trimethylsiloxy)-	C_18_H_37_NO_3_Si_3_	399.74	12.169	30800
9	Ethyl tridecanoate	C_15_H_30_O_2_	242.3975	14.205	244709
10	Undecanoic acid, ethyl ester	C_13_H_26_O_2_	214.3443	14.205	244709
11	Hexadecanoic acid, ethyl ester	C_18_H_36_O_2_	284.4772	14.205	244709
12	Decanoic acid, ethyl ester	C_12_H_24_O_2_	200.3178	14.205	244709
13	Tridecanoic acid, 12-methyl-, methyl ester	C_15_H_30_O_2_	242.3975	15.534	580270
14	Methyl stearate	C_19_H_38_O_2_	298.5038	15.534	580270
15	Docosanoic acid, ethyl ester	C_24_H_48_O_2_	368.6367	16.144	10315
16	Hexadecanoic acid, ethyl ester	C_18_H_36_O_2_	284.4772	16.144	10315
17	Trisiloxane, 1,1,1,5,5,5-hexamethyl-3,3-*bis*[(trimethylsilyl)oxy]-	C_12_H_36_O_4_Si_5_	384.8393	16.989	77570
18	Phloroglucitol	C_6_H_6_O_3_	126.11	16.989	77570
19	3-Trifluoroacetoxydodecane	C_14_H_25_F_3_O_2_	282.34	17.245	40356
20	Cyclononasiloxane, octadecamethyl-	C_18_H_54_O_9_Si_9_	667.3855	17.701	570902
21	Hexadecanoic acid, 2-hydroxy-1-(hydroxymethyl) ethyl ester	C_19_H_38_O_4_	330.5026	20.223	402651
22	Glycerol 1-palmitate	C_19_H_38_O_4_	330.5026	20.223	402651
23	Pentadecanoic acid, 2-hydroxy-1-(hydroxymet			20.223	402651
24	2-Hydroxy-1-(hydroxymethyl) ethyl icosanoate	C_23_H_46_O_4_	386.60	20.223	402651
25	Hexadecanoic acid,1,1′-[(1S)-1-(hydroxymethyl)-1,2-ethanediyl] ester	C_35_H_68_O_5_	568.91	20.223	402651
26	Octadecanoic acid, 2-hydroxy-1,3-propanediyl ester	C_39_H_76_O_5_	625.032	20.223	402651
27	10-Undecenoic acid, octyl ester	C_19_H_36_O_2_	296.4879	20.605	200047
28	Octadecanoic acid, 2-hydroxy-1,3-propanediyl ester	C_39_H_76_O_5_	625.032	20.223	159887
29	2-Hydroxy-1-(hydroxymethyl) ethyl icosanoate	C_23_H_46_O_4_	386.60	23.863	159887
30	Heptadecanoic acid, heptadecyl ester	C_34_H_68_O_2_	508.90	23.863	159887
31	13-Docosenamide, (Z)-	C_22_H_43_NO	337.5829	24.618	13399817

**Table 2 pharmaceuticals-15-01466-t002:** Comprehensive antioxidative effects of ME_X_LS.

Sample Name	TFC (mg Rutin Equivalent/g Dry Extract	TPC (mg GAE/g Dry Extract)	TAC (mg AAE/g Extract)	TPACC (mg Cat/g Dry Extract)	DPPH Free Radical Scavenging Activity (IC_50_ μg/mL)		Iron Chelating Activity (IC_50_ μg/mL)		Nitric Oxide Scavenging Activity (IC_50_ μg/mL)		Hydroxyl Radical Scavenging Activity (IC50 μg/mL)		Lipid Peroxidation Inhibition Capacity (IC_50_ μg/mL)		Protein Denaturation Inhibition Effect (IC_50_ μg/mL)	
ME_X_LS	154.06 ± 0.62	277.50 ± 2.25	157.70 ± 2.60	337.50 ± 29.92	STD (AA)	MELS	STD (AA)	ME_X_LS	STD (Quer)	ME_X_LS	STD (Cat)	ME_X_LS	STD (Cat)	ME_X_LS	STD (Asp)	ME_X_LS
					9.22 ± 0.80	337.50 ± 29.92	48.39 ± 1.87	118.48 ± 2.84	3.06 ± 0.64	14.4 ± 0.17	163.87 ± 3.35	184.40 ± 0.71	43.82 ± 3.13	60.71 ± 4.24	110.56 ± 5.23	386.44 ± 6.32

Here, total flavonoid content (TFC), total phenolic content (TPC), total antioxidant capacity (TAC), and total proanthocynidin content (TPACC) of ME_X_LS were expressed as mg/g of dry weight. AA—ascorbic acid, Quer—quercetin, Cat—catechin; Asp—aspirin were used as standard in the tabulated experiments. All the IC_50_ values of ME_X_LS are expressed as (μg/mL). Values are presented as mean ± SD.

**Table 3 pharmaceuticals-15-01466-t003:** Effect of ME_X_LS on liver, kidney, and pancreas weights in intervention period.

Groups	Liver Weight ± SD (mg)	Kidney Weight ± SD (mg)	Pancreatic Weight ± SD (mg)
NC	9.32 ± 0.05 ***	1.68 ± 0.02 **	0.67 ± 0.01 ***
RC	8.53 ± 0.09 ***	1.71 ± 0.02 *	0.47 ± 0.01 **
ME_X_LS50	9.52 ± 0.02 **	1.51 ± 0.03 ***	0.47 ± 0.00 **
ME_X_LS100	8.28 ± 0.07 ***	1.62 ± 0.02 ***	0.45 ± 0.00 *
ME_X_LS200	9.88 ± 0.04 *	1.67 ± 0.02 **	0.49 ± 0.00 ***
DC	7.34 ± 0.13	1.81 ± 0.02	0.40 ± 0.01

Kidney, liver, and pancreas weights for intervention of ME_X_LS in Albino rats over four weeks at certain temperature and pressure (*n* = 6). Data are expressed as mean ± SD. Data were analyzed by one way analysis of variance (ANOVA) using statistical software SPSS followed by a Tukey’s post hoc test. Astarisk values indicate the level of significance among the tested groups at *p* < 0.05.

**Table 4 pharmaceuticals-15-01466-t004:** Effects of ME_X_LS on different serum parameters and liver glycogen.

Parameters	NC	RC	ME_X_LS50	ME_X_LS100	ME_X_LS200	DC
ALT (U/L)	73.50 ± 1.36 ***	48.30 ± 1.66 ***	60.60 ± 2.18 ***	83.50 ± 2.01 ***	87.70 ± 1.50 ***	97.30 ± 1.86
AST (U/L)	75.20 1.85 **	85.30 ± 1.44 ***	100.60 ± 2.14 ***	122.00 ± 3.48 ***	142.00 ± 2.80 ***	170.00 ± 2.24
Creatinine (mg/dL)	0.52 ± 0.02 ***	0.55 ± 0.02 ***	0.60 ± 0.01 ***	0.45 ± 0.01 ***	0.59 ± 0.01 ***	0.90 ± 0.02
CKMB (U/L)	171.20 ± 4.12 ***	87.60 ± 3.91 ***	101.20 ± 2.23 ***	152.00 ± 4.84 ***	172.40 ± 3.14 ***	220.80 ± 2.40
HDL (mg/dL)	13.80 ± 0.37 *	16.60 ± 0.50 ***	13.60 ± 0.47 *	14.00 ± 0.71 *	17.40 ± 0.51 ***	11.60 ± 0.51
LDL (mg/dL)	47.60 ± 1.47 **	37.60 ± 1.03 ***	38.80 ± 0.86 ***	44.00 ± 1.01 ***	57.80 ± 0.86 ***	52.00 ± 1.02
LDH (U/L)	547.60 ± 5.23 ***	539.00 ± 0.41 ***	865.00 ± 8.82 ***	331.40 ± 6.01 ***	273.40 ± 3.08 ***	782.60 ± 5.09
Liver glycogen (mg/g)	0.25 ± 0.00 ***	0.30 ± 0.00 **	0.03 ± 0.00 **	0.60 ± 0.00 ***	0.054 ± 0.00 *	0.41 ± 0.01
Total cholesterol (mg/dL)	62.40 ± 2.11 ***	69.60 ± 1.08 *	54.80 ± 2.22 ***	60.80 ± 1.50 ***	67.20 ± 1.57 **	76.00 ± 0.70
TG (mg/dL)	111.00 ± 4.14 ***	42.20 ± 1.88 ***	63.80 ± 2.42 ***	117.60 ± 1.91 ***	94.40 ± 3.54 ***	167.80 ± 2.96
Urea (mg/dL)	47.00 ± 1.20 *	45.20 ± 1.30 **	37.60 ± 1.40 ***	42.10 ± 2.80 ***	45.00 ± 1.90 **	56.20 ± 2.70
Uric acid (mg/dL)	6.17 ± 0.05 **	5.88 ± 0.13 ***	5.93 ± 0.07 ***	5.05 ± 0.07 ***	5.19 ± 0.06 ***	6.46 ± 0.03
Serum Insulin level (mL U/mL)	0.12 ± 0.05 *	0.21 ± 0.06 **	0.20 ± 0.04 **	0.15 ± 0.09 *	0.11 ± 0.06 *	0.06 ± 0.02

Effect of ME_X_LS on serum AlT, AST, creatinine, CKMB, HDL, LDL, LDH, liver glycogen, total cholesterol, triglycerides, urea and uric acid and insulin in Albino rats in a four-week intervention (*n* = 6). The data is shown as Mean ± SD. Data were analyzed by one way analysis of variance (ANOVA) using the statistical software SPSS followed by Tukey’s post-hoc test. Asterisk values indicate the level of significance among the tested groups at *p* < 0.05.

**Table 5 pharmaceuticals-15-01466-t005:** Scoring for the effect of ME_X_LS on the pancreatic architectures.

Changes in Pancreatic Tissues						
	NC	DC	RC	ME_X_LS50	ME_X_LS100	ME_X_LS200
Diameter of islet of Langerhans (μm)	173 ± 47	ND	125 ± 28	220 ± 18	205 ± 13	200 ± 57
Area occupied by β-cell/islet of ± Langerhans (μm^2^)	20,703 ± 4730	ND	11,227 ± 2309	46,400 ± 1861	30,450 ± 1366	43,700 ± 5773
Necrotic cells	−	+++	+	+	−	−
Degenerated Cells	−	+++	+	+	+	+

Scoring for the histopathological assessments of pancreatic tissues are graded as follows: (−) indicates “No abnormality” (+) indicates “Mild injury” (++) indicates “Moderate injury” (+++) indicates “Severe injury”.

**Table 6 pharmaceuticals-15-01466-t006:** Scoring for the effect of ME_X_LS on the architectures of kidney tissues.

Predisposing markers	NC	DC	RC	ME_X_LS50	ME_X_LS100	ME_X_LS200
Tubular epithelial cell degeneration	−	+	−	−	−	−
Tubular epithelial cell necrosis	−	+	−	+	−	−
Increased fibrous tissue	−	+	−	+	+	−
Interstitial mononuclear cell titration	−	+	+	−	+	−
Hyperemic vessels in the interstitium	−	+	−	+	+	−
Eosinophilic secretion in the tubules lumen	−	+	−	−	−	−
Atrophic glomerulus and tubules	−	++	−	+	−	−

Plus (+) and minus (−) signs indicate the presence and absence of the predisposing markers.

**Table 7 pharmaceuticals-15-01466-t007:** Docking score for major biometabolites of ME_X_LS.

Compounds	PubChem ID	3G9E	4CFH	1PPI
Methyl alpha-d-galactopyranoside	76935	**−5.218**	**−5.764**	**−5.615**
Undecanoic acid ethyl ester	12327	−1.831	−0.509	−1.073
Hexadecanoic acid, 2-hydroxy-1-(hydroxymethyl) ethyl ester	123409	**−5.562**	**−4.791**	**−3.766**
Glycerol 1-palmitate	14900	−5.783	−1.474	−3.476
Pentadecanoic acid, 2-hydroxy-1-(hydroxymethyl)	537297	**−5.683**	**−4.824**	**−4.264**
2-Hydroxy-1-(hydroxymethyl) ethyl icosanoate	537294	**−5.372**	**−4.906**	**−3.429**
Hexadecanoic acid, 1,1′-[(1S)-1-(hydroxymethyl)-1,2-ethanediyl] ester	644078	−4.975	−2.53	−4.975
13-Docosenamide. (Z)-	5365371	**−6.785**	**−5.451**	**−3.726**
9-Octadecenamide. (Z)-	5283387	−3.059	−2.018	0.104
Decanoic acid, ethyl ester	8048	−1.372	−0.3	−1.041

PPARγ (PDB ID 3G9E); AMPK (PDB ID: 4CFH) and α-amylase enzyme (PDB ID: 1PPI); Docking scores in Kcal/mol; bold text indicates the highest score.

**Table 8 pharmaceuticals-15-01466-t008:** Physiochemical properties of the selected compounds in ME_X_LS.

Compounds	Lipinski Rules				Lipinski’s Violations	Veber Rules	
	MW (g/mol)	HBA	HBD	Log P		nRB	TPSA
Methyl α-d-Galactopyranoside	194.1825	6	4	−2.40	0	2	99.38 Å^2^
Ethyl tridecanoate	242.3975	2	0	3.94	0	13	26.30 Å^2^
Undecanoic acid. ethyl ester	214.3443	2	9	3.42	0	11	26.30 Å^2^
Decanoic acid. ethyl ester	200.3178	2	0	3.15	0	10	26.30 Å^2^
Hexadecanoic acid. 2-hydroxy-1-(hydroxymethyl) ethyl ester	330.5026	4	2	3.18	0	18	66.76 Å^2^
Glycerol 1-palmitate	330.5026	4	2	3.18	0	18	66.76 Å^2^
Pentadecanoic acid. 2-hydroxy-1-(hydroxymethyl)	316.48	4	2	2.94	0	17	66.76 Å^2^
2-Hydroxy-1-(hydroxymethyl) ethyl icosanoate	386.60	4	2	4.06	0	22	66.76 Å^2^
Hexadecanoic acid.1.1′-[(1S)-1-(hydroxymethyl)-1.2-ethanediyl] ester	568.91	5	1	6.27	2	34	72.83 Å^2^
13-Docosenamide. (Z)-	337.5829	1	1	5.06	1	19	43.09 Å^2^
9-Octadecenamide. (Z)-	281.4766	1	1	4.16	1	15	43.09 Å^2^

MW, molecular weight (≤500 g/mol); HBA, hydrogen bond acceptor (≤10); HBD, hydrogen bond donor (≤5); Log P, lipophilicity (≤4.15); nRB: number of the rotatable bond (≤10); TPSA: topological polar surface area (≤140).

## Data Availability

Data is contained within the article or Appendix A.

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
