# Peer review of "Natural Compounds of Lasia spinosa (L.) Stem Potentiate Antidiabetic Actions by Regulating Diabetes and Diabetes-Related Biochemical and Cellular Indexes†"

_pharmaceuticals, 2022, doi:10.3390/ph15121466_

Round 1

Reviewer 1 Report (Previous Reviewer 2)

The main concept of the article is interesting. The experimental part is extensive. Discussion over the results of the experiments and the conclusions are proper. Nonetheless, to be acceptable for publication, the paper requires the following improvements: 

1) The paper should be prepared in line with the requirements of the Journal including giving text into the template as well as adequate preparing of all sections (including e.g. section References). 

2) Abstract of the paper should be significantly shortened.

3) The paper needs to be written in the passive voice instead of the active one (e.g. phrases as "we have been reported..." should be replaced by "it has been reported..." etc.).

4) In the case of some applied reagents some more information should be given in section Materials, e.g. the origin of starch and albumin should be added.

5) Section 2.5.1.: the main principles of the procedures applied aiming at determining the TPC, TF, TAC and TPACC  should be clearly indicated.

6)  Second paragraph of section 3.1.: the retention times have been given without units.

7) Authors mentioned that "Several native medicinal herbs have a strong potential for suppressing the activity of α-amylase..." - some examples should be provided.

8) Tables 3-4.: the values and the standard deviations should have the same number of significant digits after a decimal point.

9) Figure captions should be shortened. For example, the name of the apparatus used should be given in the description of the methodology applied, not in a figure caption.

10) Figure 1. is of a very poor quality, it is difficult to notice the data.

11) Paper should be re-checked grammatically and stylistically. Next, it contains some misspellings (e.g. "diabtes" instead of "diabetes") which should be corrected.

Author Response

Reviewer 2 Report (Previous Reviewer 1)

Accept in present form

Author Response

Reviewer 3 Report (New Reviewer)

1. The graphical abstract is very messy. It should be simplified.

2. Page 7 Section 2.2 Adequate Reference should be provided.

3. Page 8 Line 15 the symbol of degree should be corrected.

4. Page 9 Line 6 symbol should be used for percent in place of text.

5. Page 9 Line 20 (r.t.) stands for what? If room temperature then mention the actual temperature of the room.

6. Page 11 Line 22, (50 Mm) should be written as 50 mM.

7. Page 12 Line 5, the reference (Saeed et al. 2012) should be given a Reference Number.

8. Page 12 Line 10 the sentence “The membrane-stabilizing activity was determined by the method” seems to be incomplete.

9. Page 14 Line 12 Specify the reason for the selection of male adult albino rats of the Wistar strain only. Why not female rats or a combination of male and female rats?

10. Page 14 Line 18 Date of animal protocol approval is missing.

11. Page 15 Line 12 For the induction of diabetes single injection of streptozotocin was given or multiple injections were given. Kindly mention it.

12. Page 16 Line 14, Kindly correct the organ storage temperature. It is -20 °C.  

13. Page 18 Line 13, the reference (Hussein et al., 2015) should be given Reference Number.

14. Page 18 Line 16, the sentence “In Hossen et al., a brief description of the molecular docking study is provided [39].” Should be corrected.

15. Page 33 Section References, the references Saeed et al. 2012, Hussein et al., 2015 are missing.

16. Ref 18 should be corrected as it is not complete. Check the spelling also.

17. Kindly check the uniformity in the presentation of the concentration of reagents throughout the manuscript.

18. All the references must be presented in a uniform style.

19. Kindly check the footnote of Table 8 and correct it.

20. Page 51 Line 3, present the microscopic resolution as 10 X 40.

21. The language should be more refined.

Round 2

Reviewer 1 Report (Previous Reviewer 2)

The paper has been corrected in line with the recommendations therefore it may be accepted for publication in Pharmaceuticals.

This manuscript is a resubmission of an earlier submission. The following is a list of the peer review reports and author responses from that submission.

Round 1

Reviewer 1 Report

Md. Mamunur Rashid et al., Natural compounds of Lasia spinosa (L.) stem potentiate antidiabetic actions by regulating diabetes and diabetes-related biochemical and cellular indexes. However, the English grammatical errors detract from the paper's quality, and hence it needs to be improved. Furthermore, the authors need to address the following major and minor concerns.

- A good abstract must have the followings: Motivation and problem statements. Brief description of the methods. Important results and results that supported them. Concluding remark.

-Introduction: Establishment of current knowledge of the field;  Summarize previous research, providing the wider context and background of the importance of the current study; Set the stage for the present research, indicating gaps in knowledge and presenting the research question. The author should be included “stand drug side effects, and plant benefit of antidiabetic activities (update reference).

-3.1. Chemicals and reagents the chemicals and reagents were ensured as analytical grade until. What do you mean? Do you mean of analytical grade?

-Any specific reasons for choosing male albino rats? Why female rats were not included?

-Fasting blood glucose levels of >300 mg/dL were considered as diabetic…, which glucose kit was used? Company name?

-How to animal Sacrifice …author should explain anesthetized agent. Please justify, what was dose of anesthetized alone did you use.

-How selected dose of MEXLS 50, 100, 200 mg/kg bw?

Did the authors use only one statistical test?

-Detail interactions in docked complexes should be provided in the text  and discuss their role in the protein structure with suitable references.

-Overall, the “Discussion” section needs to be revised and provide more comparative study than just mentioning the obtained results.

Conclusion: please add “future prospects”.

-In reference section according to the Journal format. The author should mention the genus and the species name should be in italics.

Author Response

Dear Editor

Thank you so much for your effort to have the manuscript reviewed. We also would like to thank the reviewer for their efforts in reviewing the manuscript. We have gone through the comments of the reviewers and addressed them in a point-to-point manner. Enclosed please find the author’s responses against reviewers’ queries:

Md. Mamunur Rashid et al., Natural compounds of Lasia spinosa (L.) stem potentiate antidiabetic actions by regulating diabetes and diabetes-related biochemical and cellular indexes. However, the English grammatical errors detract from the paper's quality, and hence it needs to be improved. Furthermore, the authors need to address the following major and minor concerns.

- A good abstract must have the followings: Motivation and problem statements. Brief description of the methods. Important results and results that supported them. Concluding remark.

Author’s Response: Thank you very much for your valuable suggestion. The Abstract has been revised and abstract through incorporating the suggestions.

-Introduction: Establishment of current knowledge of the field;  Summarize previous research, providing the wider context and background of the importance of the current study; Set the stage for the present research, indicating gaps in knowledge and presenting the research question. The author should be included “stand drug side effects, and plant benefit of antidiabetic activities (update reference).

Author’s Response: Dear reviewer, thank you for your very constructive suggestions. The introduction part has been improved in the context of your suggestion.

-3.1. Chemicals and reagents the chemicals and reagents were ensured as analytical grade until. What do you mean? Do you mean of analytical grade?

Author’s Response: Thank you so much for your question. Yes, most of the chemicals and reagents were from analytical grade, however, few of the chemicals are of other grades such as HPLC grade or reagent grade which are specified in respective cases.

-Any specific reasons for choosing male albino rats? Why female rats were not included?

Author’s Response:  The reason the researcher avoids female subjects is because of their hormone fluctuation during the reproductive cycle. This hormone fluctuation may influence your results. You do not have this problem with male rats because they have a more stable hormonal status. Furthermore, male rats tend to develop more pronounced insulin resistance whilst females show a greater loss of insulin release and beta cell mass. Therefore. Most of the studies are done with males. Importantly, the female rats had lower survival rates than males, which would make the males more attractive for longer-term drug study design.

-Fasting blood glucose levels of >300 mg/dL were considered as diabetic…, which glucose kit was used? Company name?

Author’s Response: Fasting blood glucose was measured by using a glucometer (Glucoplus (Glucoplus Inc., Saint- Laurent, Quebec, Canada) via tail prick method.

-How to animal Sacrifice …author should explain the anesthetized agent. Please justify, what a dose of anesthetized alone did you use.

Author’s Response: Thank you so much. We already mentioned that halothane anesthesia was used. For anesthesia 1-3% halothane in air, O2 is prescribed for animal anesthesia. We used 1% halothane.

-How selected dose of MEXLS 50, 100, 200 mg/kg bw?

Author’s Response: Thank you so much for your insightful question. We have selected the dose from an acute toxicity study measuring the LD50 value. To detail, a single dose of extract has been increased to administrate until animals show toxicity. The 1/10 of the highest nontoxic dose is termed the optimum dose which is geometrically adjusted using a lower and higher dose to optimize the errors in the biological system (Zaoui et al., 2002)

Did the authors use only one statistical test?

Author’s Response: So far, the question we understand, we have used both GraphPad Prism and SPSS (statistical package for social science) for statistical analyses. However, Data analyses were done by One Way Analysis of Variance (ANOVA) using SPSS followed by Tukey’s multiple tests. ANOVA gives the relation between the groups and among the groups. It is more than enough for such data analysis.

-Detail interactions in docked complexes should be provided in the text and discuss their role in the protein structure with suitable references.

Author’s Response: Thank you so much for your suggestion. We have revised and extended our discussion to address the suggestion.

-Overall, the “Discussion” section needs to be revised and provide more comparative study than just mentioning the obtained results.

Author’s Response: The discussion has been improved and more insights have been added.

Conclusion: please add “future prospects”.

 Author’s Response: Future prospects of the work have been added.

-In the reference section according to the Journal format. The author should mention the genus and the species name should be in italics.

Author’s Response: References have been reformatted based on the guidelines of the Journal. Genus and species names are changed into italics.

Reviewer 2 Report

The research topic seems to be interesting. Nonetheless, the paper requires many corrections, mainly in terms of the editorial aspect. The paper should be re-checked and prepared according to the requirements of the Journal. All comments are given in more detail below:

1) The notation of the references needs to be applied in line with the requirements of the Journal (i.e. a number in brackets, not a surname and a year).

2) The paper needs to be written in a passive voice, not in an active one (phrases as "we looked into..." (line 78) need to be removed).

3) Line 132: a doble space should be removed "(...) respectively.  The (...)". The same applies to the lines 133, 154, 162 etc. Additionally, this paragraph ends with two dots.

4) The notation of sodium phosphate should be improved (line 225).

5) What means "normal temperature"? (line 237)

6) The name of section 4. - the dot should be removed.

7) Final conclusions as well as the abstract of the paper should be supplemented with more quantified data.

8) Section References should be prepared in line with the requirements of the Journal. 

9) Tables 2, 4 and 8 are not fully visible.

10) Figure captions need to be shortened. For example, information concerning the names of the apparatus should be placed in description of applied methodologies, not in figure captions.

11) Figure 1. should be improved - now it is poorly visible.

12) Figure 2: description of y axis - it should be "concentration" instead of "cocentration", additionally this description should be inverted.

13) Figure 4: What means "a, b, c, d and e" visible in this figure?  

Author Response

Review 2

Dear Editor

Thank you so much for your effort to have the manuscript reviewed. We also would like to thank the reviewers for their efforts in reviewing the manuscript. We have gone through the comments of the reviewers and addressed them in a point-to-point manner. Enclosed please find the author’s responses against reviewers’ queries:

The research topic seems to be interesting. Nonetheless, the paper requires many corrections, mainly in terms of the editorial aspect. The paper should be re-checked and prepared according to the requirements of the Journal. All comments are given in more detail below:

1) The notation of the references needs to be applied in line with the requirements of the Journal (i.e. a number in brackets, not a surname and a year).

Author’s Response: Thank you so much for your suggestion. References and citations have been reformatted in accordance with the guidelines of the Journal.

2) The paper needs to be written in a passive voice, not in an active one (phrases as "we looked into..." (line 78) need to be removed).

Author’s Response: Thank you so much for your keen observation. The manuscript has been rechecked to avoid the active voice. 

3) Line 132: a double space should be removed "(...) respectively.  The (...)". The same applies to the lines 133, 154, 162 etc. Additionally, this paragraph ends with two dots.

Author’s Response: Suggested issues have been corrected.

4) The notation of sodium phosphate should be improved (line 225).

Author’s Response: Corrected.

5) What means "normal temperature"? (line 237)

Author’s Response: Normal temperature refers to the room temperature.

6) The name of section 4. - the dot should be removed.

Author’s Response: The dot has been removed.

7) Final conclusions as well as the abstract of the paper should be supplemented with more quantified data.

Author’s Response: Both the abstract and conclusions have been revised in line with the suggestion.

8) Section References should be prepared in line with the requirements of the Journal. 

Author’s Response: References have been reformatted based on the guidelines of the Journal. Genus and species names are changed into italics.

9) Tables 2, 4 and 8 are not fully visible.

Author’s Response: I am sorry for the inconvenience. We can clearly visualize the tables in the file we have. I would rather request the editorial team to assist in this issue.

10) Figure captions need to be shortened. For example, information concerning the names of the apparatus should be placed in description of applied methodologies, not in figure captions.

Author’s Response: Thank you for your important suggestion. We have added some description of the GC-MS device to make the figure legends self-explanatory which is always desired. However, we have shortened the legend/captions of the figure.

11) Figure 1. should be improved - now it is poorly visible.

Author’s Response: The resolution of Figure 1 has been improved.

12) Figure 2: description of y axis - it should be "concentration" instead of "cocentration", additionally this description should be inverted.

Author’s Response: The typo mistake has been corrected.

13) Figure 4: What means "a, b, c, d and e" visible in this figure?  

Author’s Response: Significant differences between and among the groups are affirmed by the superscript letters over the bar graph. The same letter does mean they don’t have any significant difference.

Reviewer 3 Report

Lasia spinosa is a herb that grows in Asia and may have medical effects. As an extent of their previous study, the authors tried to discover the anti-diabetic effect of L. spinosa methanol stem extract (MExLS). This work is very intriguing and you have made a great effort, however, I have a few suggestions to make regarding the manuscript: 

Major issues:

1, Why choose Lasia spinosa for diabetes? No previous study or ancient medical books mentioned this anti-diabetic effect of Lasia spinosa in your Introduction part.

2, As you mentioned in the Introduction part, the antioxidant effect of Lasia spinosa had been reported in previous studies, why did you repeat this result in your study?

3, Line 94, you mentioned in vitro model, however, no result yield in in vitro model was present in your study.

4, Line 97, there is no Table S1.

5, The information in some figures or tables is missing (e.g. Table 2, Table 8, and Figure 6).

6, As you mentioned in 3.2, the MExLS was semisolid. How to feed rats with MExLS, dilute them in water/organic solvent or mix them in chow? 200 mg/kg is not a small amount for a rat.

7, The two different things had the same abbreviation, which may confuse the reader (e.g. DPPH in Line 294 and Line 297; PDB in Line 479 and Line 481).

8, In 3.6.3 you used metformin as a reference control, where did you purchase this product? Is it merchandise or raw material? How to decide the dosage?

9, In 3.8.1, why did you choose PPAR, AMPK, and PDB for molecular docking? What are the protein expression of PPAR, AMPK, and PDB in rats after 4-week-treatment?

10, The meaning of two sentences with Line 425 to Line 428 was the same, please remove one of them.

11, As you mentioned insulin and inflammation in the Discussion part, however, you did not measure these indexes (e.g. insulin, IL-6, TNF-α) in your animal model.

12, You also mentioned the adverse effects of diabetes in the liver and heart, however, what is the tissue architecture like in these organs?

Minor issues:

1, The expressions were not unified throughout the manuscript, e.g. MExLS and MEXLS, IC50 and IC50, p and P, r.t., RT and room temperature.

2, Maybe the symbol < is missing in Line 130. Line 284, 0.084.

3, There are some typos, e.g. standra (Line 281), boy weights (Line 426).

In my opinion, the result of the antioxidant effect should be removed. After the result of GC-MS, the authors should predict the effect relevant to diabetes of MExLS by computational studies, then prove those findings in the animal model. Besides, how the authors treated the manuscript (layout, typo, different typeface, corner mark) may reflect their attitude toward the study, which means unsatisfactory.

Author Response

Review 3

Dear Editor

Thank you so much for your effort to have the manuscript reviewed. We also would like to thank the reviewers for their efforts in reviewing the manuscript. We have gone through the comments of the reviewers and addressed them in a point-to-point manner. Enclosed please find the author’s responses to reviewers’ queries:

Major issues:

1, Why choose Lasia spinosa for diabetes? No previous study or ancient medical books mentioned this anti-diabetic effect of Lasia spinosa in your Introduction part.

Author’s Response: Thank you so much for your insightful question. The plant has been reported for its glucose-lowering effect which is mistakenly missing in our introduction section but fortunately, those literatures helped us carry on the current research.

Hasan MN, Munshi M, Rahman MH, Alam SMN, Hiroshima A. Evaluation of antihyperglycemic activity of Lasia spinosa leaf extracts in swiss albino mice world journal of pharmacy and pharmaceutical sciences volume 3, issue 10, 118-124.

Tran Thanh Men, Pham Ngoc Khang, Truong Thi Phuong Thao, Do Tan Khang, Luu Thai Danh, Nguyen Trong Tuan and Dai Thi Xuan Trang, 2021. Phytochemical Screening and Antioxidant, Anti-diabetic Properties Evaluation of Lasia spinosa L. Thwaites Stem Extracts. Asian Journal of Plant Sciences, 20: 571-577.

2, As you mentioned in the Introduction part, the antioxidant effect of Lasia spinosa had been reported in previous studies, why did you repeat this result in your study?

Author’s Response: Very intuitive question indeed. Let us make the answer a bit longer. You might be agreed that the extent of the same biological function of plant parts may be different because of the difference in phytochemical nature and quantity. Similarly, the nature of the solvent used for plant material extraction guides the nature and quantity of the same biological function in the same part of the plant. The reported antioxidative effect was done with Ethyl acetate, hexane, and ethanol extract. Additionally, previous research was done on the leaf of Lasia spinosa. Whereas, in our research, we are reporting the methanol extract of Lasia spinosa stem which will make a complete antioxidative profile of this plant, especially the stem parts.  Therefore, it is not repetition in fact.

3, Line 94, you mentioned in vitro model, however, no result yield in in vitro model was present in your study.

Author’s Response: Thank you so much. We have written the quantitative value of yield 12.0 g in the collection and extract of plant material section. By the way, yield % is incorporated.

4, Line 97, there is no Table S1.

Author’s Response: We are sorry, that Table S1 was missing. We have added Table S1.

5, The information in some figures or tables is missing (e.g. Table 2, Table 8, and Figure 6).

Author’s Response: Thank you so much. We got something missing in Table 2 and Figure 6 but not in 8. It would be appreciable if you please specify.

6, As you mentioned in 3.2, the MExLS was semisolid. How to feed rats with MExLS, dilute them in water/organic solvent or mix them in chow? 200 mg/kg is not a small amount for a rat.

Author’s Response: True it was semisolid and dissolved in water to feed using a feeding needle. Well, the dose of 200 mg/kg body weight means that the animal which has a body weight of 180 g needs 36 mg. Such an amount is easy to feed.

7, The two different things had the same abbreviation, which may confuse the reader (e.g. DPPH in Line 294 and Line 297; PDB in Line 479 and Line 481).

Author’s Response: Dear reviewer, thank you for your observation. But they are not the same thing. DPPH means 2,2-diphenyl 1-picryl hydrazyl or you can say 1,1,-diphenyl 2-picry hydrazyl. But PDB stand for Protein Data Bank which preserves the detail information for all the proteins like gene data bank. We have used three receptor proteins peroxisome proliferator-activated receptor-gamma (PPARγ, PDB ID 3G9E), AMP-activated protein kinase (AMPK, PDB ID: 4CFH), and α-amylase enzyme (PDB ID: 1PPI) for interactions. By the way, we have used the full form first in the abstract section.

8, In 3.6.3 you used metformin as a reference control, where did you purchase this product? Is it merchandise or raw material? How to decide the dosage?

Author’s Response: This is kindly donated as rea material by the leading pharmaceutical manufacturing company in our country, the SQUARE Pharmaceuticals Ltd. The dose is literally established. The information is given in the abbreviation section. However, we have added the information in chemicals and reagent section as well.

9, In 3.8.1, why did you choose PPAR, AMPK, and PDB for molecular docking? What are the protein expression of PPAR, AMPK, and PDB in rats after 4-week-treatment?

Author’s Response: Thank you for your query. Actually we used three receptor proteins peroxisome proliferator-activated receptor-gamma (PPARγ), AMP-activated protein kinase (AMPK) and, and α-amylase enzyme receptor. For drug discovery, in silico/computational studies use these receptor proteins to find out the drug-likeliness of new molecules isolated or identified. And PDB means the Protein Data Bank (https://www.rcsb.org/) which preserves all the information for a protein. And every protein has a unique ID. We must use that ID to identify the protein. Therefore, we call PDB ID.

10, The meaning of two sentences with Line 425 to Line 428 was the same, please remove one of them.

Author’s Response: I think the author’s version is different from that reviewer’s one and therefore the line number is a bit different. By the way, we have adjusted the overlapping and close-meaning sentences.

11, As you mentioned insulin and inflammation in the Discussion part, however, you did not measure these indexes (e.g. insulin, IL-6, TNF-α) in your animal model.

Author’s Response: Very thoughtful question indeed. We appreciate. In the discussion part, we tried to justify how inflammation is linked with diabetes. And how antidiabetic drugs play inti-inflammatory roles. We focused that inflammation helps release the lysosomal hydrolytic enzymes which destabilize the lysosomal membranes. Therefore, membrane stabilizers are important to inhibit protein denaturation while antidiabetic drugs such as TZDs, DPP-4 inhibitors, GLP-1 RAs, and insulin contribute to this process of inhibiting protein denaturation. We connected the issue because we have evaluated the protein denaturation inhibition and lipid peroxidation inhibition capacity of MEXLS which are summarized in Table 2. We relevantly used the terminologies for interpretation.

12, You also mentioned the adverse effects of diabetes in the liver and heart, however, what is the tissue architecture like in these organs?

 Author’s Response: Well, relevant question, as we know that the secondary complications of diabetes are- diabetes nephropathy, diabetes retinopathy, diabetes cardiomyopathy, and diabetes Cerebroencepalopathy. These indicate the organs which could be affected by diabetes. Therefore, we have at least incorporated the biochemical markers for the heart and liver as well as liver glycogen is estimated (Table 4).  Another study is expected to design for observing the changes in tissue architectures for the liver and heart in diabetes.

Minor issues:

1, The expressions were not unified throughout the manuscript, e.g. MExLS and MEXLS, IC50 and IC50, p and P, r.t., RT and room temperature.

Author’s Response: We have noticed and harmonized. Thank you so much.

2, Maybe the symbol “<” is missing in Line 130. Line 284, 0.084.

Author’s Response: Corrected.

3, There are some typos, e.g. “standra” (Line 281), boy weights (Line 426).

 Author’s Response: Corrected.

In my opinion, the result of the antioxidant effect should be removed. After the result of GC-MS, the authors should predict the effect relevant to diabetes of MExLS by computational studies, then prove those findings in the animal model. Besides, how the authors treated the manuscript (layout, typo, different typeface, corner mark) may reflect their attitude toward the study, which means unsatisfactory.

Author’s Response: Thank you so much for your suggestion. I think we have explained in response to your question no 2 which implies that the antioxidative effect of Methanol stem extract is no longer done and therefore, it’s important for justification. Because oxidative stress is thought as a pivotal factor for diabetes. Strong antioxidative properties of plant extract play a significant role in the attenuation of diabetes and related complications. We hope, the reviewer will accept the use of antioxidative data from this perspective. However, if you still think to remove that part, we can reconsider. Other minor typo mistakes and syntax errors have been carefully corrected.

Round 2

Reviewer 2 Report

Authors made all suggested corrections thus in the opinion of the reviewer the manuscript may be accepted for publication.

Reviewer 3 Report

The authors tried their best to fix some mistakes, however, I have some fundamental concerns to make regarding the manuscript: 

 1, Line 896-Line 899: “In a 4 weeks intervention, fasting blood glucose, body weights, foods, and fluid intakes were recorded every week. Every week (total 4 weeks) each fasting blood glucose level and body weight were measured and recorded data. Food and fluid intake were also observed.” The first sentence has the similar meaning as the other two, please remove one of them.

2, The expressions MEXLS, IC50, and missing < still remain in figure legends.

3, Line 1127, mice?

4, The group names in Figure 5 were not shown completely, while in Figure 6 had changed another style.

I strongly recommend providing the results of insulin and inflammatory factors in rats (ELISA test), tissue architecture of the liver and heart (H&E staining), and protein expressions of PPAR, AMPK and PDB (Western Blotting).